# EveryGraph: Task-Agnostic Graph Neural Architectures

## Abstract

Graph Neural Networks (GNNs) have demonstrated remarkable performance across a range of applications. However, their generalization across diverse tasks and datasets remains a fundamental challenge. Inspired by the success of foundation models in language, vision, and audio domains, recent efforts have sought to bring similar capabilities to graph-based learning. We introduce EveryGraph, a task-agnostic GNN architecture that generalizes across more than 20 benchmark datasets spanning link prediction, node classification, and graph classification. EveryGraph introduces a novel method for constructing a unified representation space. Unlike traditional embedding techniques that may lose local context, our approach explicitly preserves the topological structure of the data, ensuring that intrinsic inter-node dependencies are accurately mapped into the latent space. Notably, EveryGraph is trained in a self-supervised regime on a single dataset and achieves this broad generalization by utilizing an in-context like downstream task adaptation.

## 1 Introduction

Graph Neural Networks (GNNs) (Kipf & Welling, 2017; Defferrard et al., 2016; Veličković et al., 2018) have emerged as a powerful tool for learning from graph-structured data, demonstrating impressive performance across a wide range of applications, including social network analysis, molecular property prediction, and recommendation systems. In recent years, the popularity of GNNs has surged, driven by advances in deep learning and the increasing availability of large-scale graph datasets.

The success of foundation models in NLP (Vaswani et al., 2017; Brown et al., 2020; Devlin et al., 2019), computer vision (Radford et al., 2021; Dosovitskiy et al., 2021), and speech recognition (Hsu et al., 2021) has spurred substantial interest in developing analogous models for graph-structured data. These foundation models, trained on large-scale and diverse datasets, have demonstrated impressive generalization capabilities across a wide range of downstream tasks. However, extending this paradigm to the graph domain introduces a unique challenge. Where text and visual data have a natural, lossless, dataset-agnostic representation space (tokens, RBG values, respectively), graph dataset differ on feature meaning and representations.

The graph deep learning community has yet to reach a consensus on what constitutes a graph foundation model (GFM). While there is an agreement that GFMs transfers from pretraining data to downstream task, different concrete approaches where suggested. Finkelshtein et al. (2025) suggest that GFMs should extent symmetry equivariance from node permutations to features and labels permutations. Zhao et al. (2025) offer a slightly more general framework: A GFM is a model pretrained on graph $G$ from feature and labels spaces $(\mathcal{X}, \mathcal{Y})$ that is able to perform inference on graph $G'$ from different feature and label spaces $(\mathcal{X}', \mathcal{Y}')$. In this work, we explore a GFM framework that extends the notion by Zhao et al. to different graph tasks: rather than focus on transfer between (for example) node-level tasks, we further explore transfer between link-, node-, and graph-level tasks.

In order to facilitate processing of data independent of the input feature space, we introduce a random projection layer. This novel approach allows the model to learn a unified representation space, independent of the input shape. Recent studies have shown that incorporating randomness can improve both the performance and expressivity of GNNs (Abboud et al., 2021; Eliasof et al., 2023), and random projections are well-studied operations. However, they weren't yet applied towards GFMs. These limitations underline

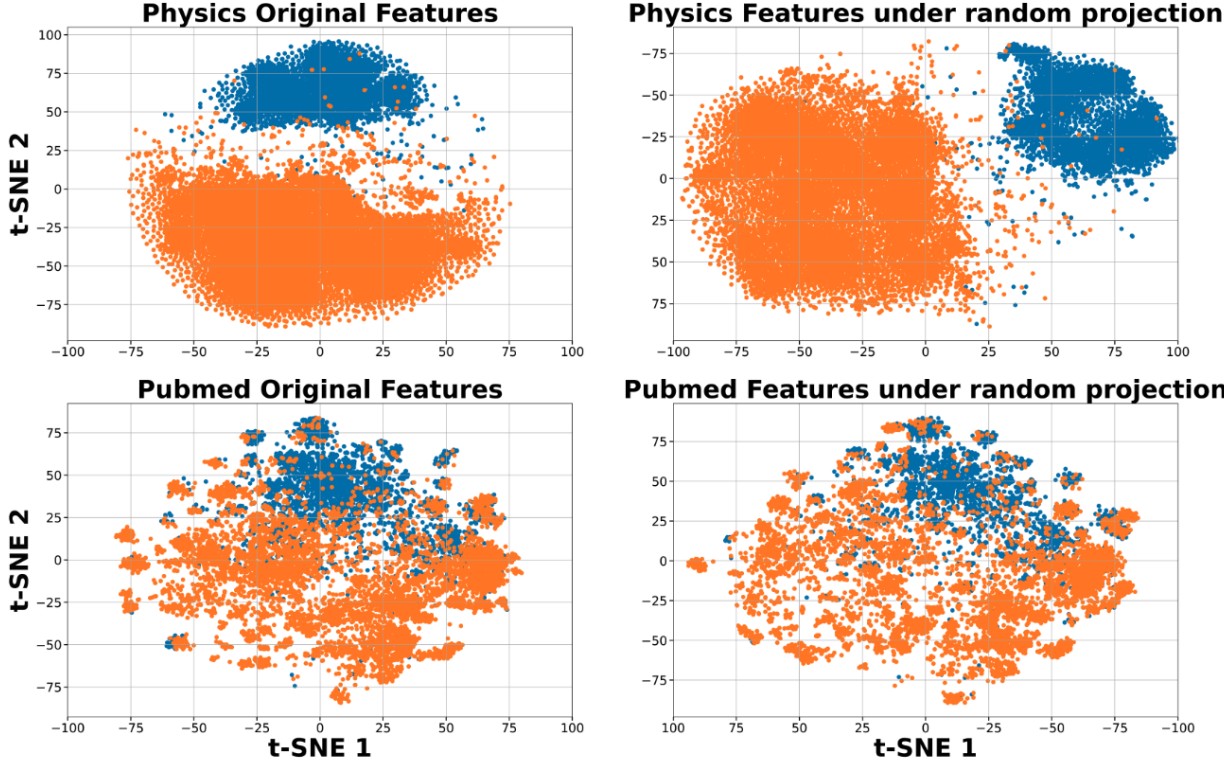

Figure 1: t-SNE visualizations for two classes. The left column shows the original t-SNE embeddings, while the right column presents the embeddings after applying a random Gaussian projection to the data. The first row corresponds to the Physics dataset, and the second row to the Pubmed dataset. For improved visual comparability, the y-axis of the Physics dataset's projected features and the x-axis for the Pubmed dataset's projected features are flipped.

| Model ↓ | Cora | | | Citeseer | | | CoCS | | |
|---|---|---|---|---|---|---|---|---|---|
| | Features | RNF | RandProj | Features | RNF | RandProj | Features | RNF | RandProj |
| **GCN** | $81.73_{\pm0.39}$ | $53.60_{\pm0.93}$ | $80.70_{\pm0.51}$ | $72.07_{\pm0.26}$ | $30.77_{\pm0.34}$ | $70.20_{\pm0.22}$ | $91.42_{\pm0.24}$ | $43.34_{\pm1.32}$ | $89.84_{\pm0.81}$ |
| **GAT** | $81.73_{\pm0.39}$ | $59.27_{\pm0.46}$ | $80.13_{\pm0.12}$ | $71.47_{\pm0.65}$ | $36.97_{\pm2.12}$ | $70.47_{\pm0.34}$ | $88.60_{\pm0.08}$ | $60.81_{\pm1.01}$ | $89.42_{\pm0.78}$ |
| **GATv2** | $81.63_{\pm0.80}$ | $59.17_{\pm1.88}$ | $80.13_{\pm0.53}$ | $71.97_{\pm0.40}$ | $37.83_{\pm2.05}$ | $70.63_{\pm0.59}$ | $88.11_{\pm1.40}$ | $59.13_{\pm2.57}$ | $89.17_{\pm0.21}$ |

Table 1: Model classification accuracy across different feature representations. Features - stands for the original graph's features also marked X. RNF - stands for random noise features also marked X". RandProj - stands for random projection features also marked as X'.

the need for more versatile and universally applicable foundation models for graph data. In Section 3.1 we provide theoretical grounds supporting random projections applicability to our goal. Figure 1 demonstrates that random projections preserve meaningful information found in the input data.

Aiming to support graph tasks across the node-, link-, and graph-levels, we propose training on link prediction task. This task allows for self-supervised learning and is independent of the pretraining datasets' designated task and label space. As discussed in Section 3 and depicted in Figure 3, we adapt pretrained representations to downstream tasks via closed-form solutions in an in-context learning framework (Brown et al., 2020).

**Our Contributions.** In this work, we introduce EveryGraph, a task-agnostic neural architecture for graph learning that demonstrates strong generalization across diverse datasets and tasks. EveryGraph is trained

on a single dataset, and successfully adapts to over 20 benchmark datasets, spanning link prediction, node classification, and graph classification. Our contributions can be summarized as follows:

- We introduce a novel, task-agnostic graph neural architecture that generalizes across diverse tasks and domains, tackling the difficulties of graph learning by utilizing theoretically grounded random projections and non-parametric in-context learning adaptation.

- We provide a theoretical and empirical exploration of random Gaussian projections as a means to develop a unified, task-agnostic representation space. Table 1 exemplifies the power of the random projection which will be elaborated, in the coming sections.

- We evaluate EveryGraph on over 20 benchmark datasets, demonstrating competitive performance with fully-trained models and superior results compared to existing in-context methods.

Our code is available at:
https://anonymous.4open.science/r/EveryGraphTaskAgnosticNeuralArchitecture-55DB/

## 2 Related Work

Graph Foundation Models (GFMs) leverage large-scale pretraining to enhance generalization across diverse downstream tasks. Inspired by foundation models in NLP (Vaswani et al., 2017; Brown et al., 2020; Devlin et al., 2019) and vision (Radford et al., 2021; Dosovitskiy et al., 2021), they show strong performance and adaptability. This section reviews advances in graph representation learning, pretraining, and transfer learning on graph-structured data.

**Graph Representation Learning.** Traditional graph learning relies on handcrafted features and spectral methods like Laplacian eigenmaps (Belkin & Niyogi, 2003) for node and graph embeddings. The advent of GNNs, including GCNs (Kipf & Welling, 2017), GATs (Veličković et al., 2018), and GraphSAGE (Hamilton et al., 2017), enabled end-to-end representation learning via message passing. However, these models often face oversmoothing and scalability issues on large graphs (Serafini & Guan, 2021). In inductive settings, GNNs struggle with dynamic changes in node features, structures, and motifs across datasets (Hamilton et al., 2017). Their task-specific training limits generalization, leading to poor transferability and substantial performance drops on unseen tasks (Anastasopoulos & Papalexakis, 2018; Rusu et al., 2016).

**Pretraining Techniques for Graphs.** Pretraining in graph learning has been widely explored through self-supervised approaches, including contrastive learning, generative modeling, and masked prediction tasks (You et al., 2020a;b). These methods aim to learn transferable representations for downstream tasks like node classification, link prediction, and graph classification. Contrastive methods (You et al., 2020b; Hassani & Khasahmadi, 2020), such as GraphCL (Zhu et al., 2021) and InfoGraph (Sun et al., 2020), maximize mutual information between augmented views of the same graph using techniques like node dropping, edge perturbation, and subgraph sampling. While effective in aligning similar structures and separating dissimilar ones, these methods depend on handcrafted augmentations that may not generalize across graph types. To address this, Bevilacqua et al. Generative approaches, such as GraphMAE (Hou et al., 2022), use masked autoencoders to reconstruct missing features or edges by learning structural and attribute-level dependencies. These models generalize well but can suffer from over-smoothing and limitations in structural representation, reducing performance on unseen graphs. Graph-based VAEs (Kipf & Welling, 2016; Simonovsky & Komodakis, 2018; Jin et al., 2018) have also been used to model latent graph distributions, aiding applications like drug discovery and molecular generation.
Bechler-Speicher et al. (Bechler-Speicher et al., 2024) argue that the lack of strong benchmarks is a key reason behind the absence of a true graph foundation model. While we share this view and have experienced the highlighted challenges, we take it as motivation to continue pursuing one.

**Transfer Learning and Multi-Task Adaptation.** GFMs aim to generalize across diverse graph tasks, including node classification, link prediction, and molecular property prediction. Multi-task frameworks like

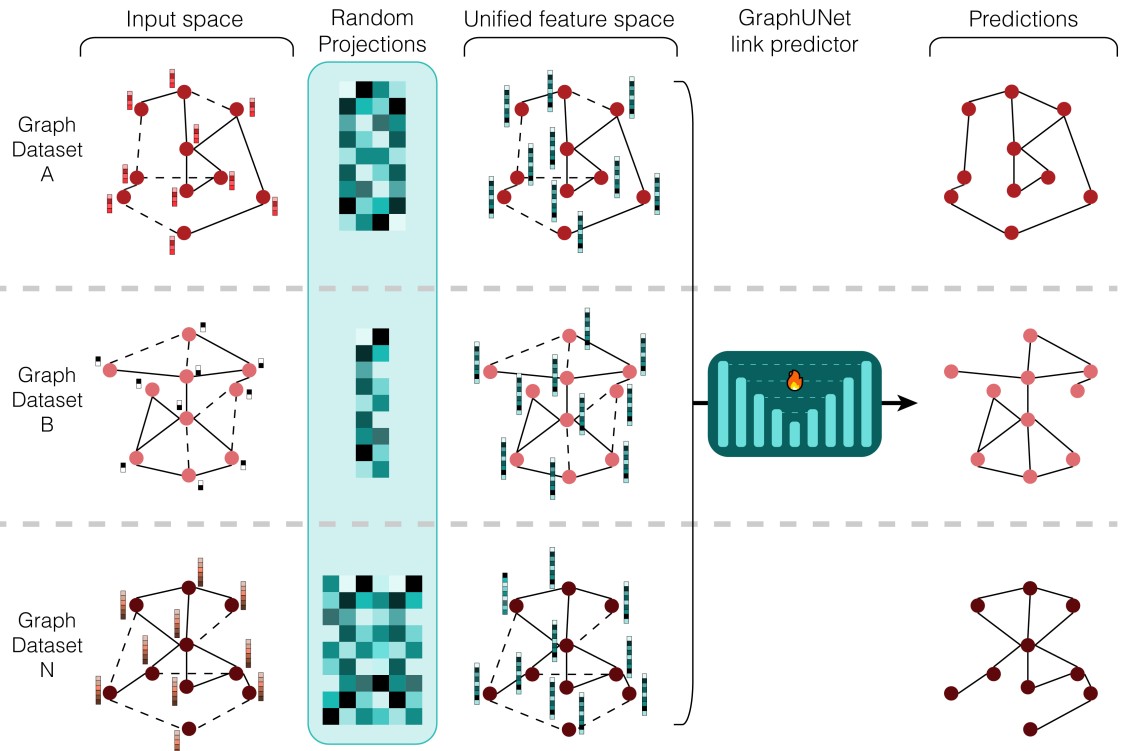

Figure 2: EveryGraph Training. EveryGraph takes any input graph dataset and embeds it into a unified representation space via random projection. A GraphUNET, trained solely on a link prediction objective, serves as the only trainable component in the pipeline.

Graph Prompting (Tian et al., 2024) and Meta-GNNs (Mandal et al., 2021) show promising cross-domain generalization. Integrating external knowledge, such as large-scale knowledge graphs, further enhances reasoning over structured data (Schlichtkrull et al., 2018; Yasunaga et al., 2021). However, these models face challenges in transfer learning and multi-task adaptation. Negative transfer may arise when task knowledge interferes across domains with differing graph structures or node distributions (Zhao et al., 2024). Multi-task learning can also lead to optimization conflicts, as tasks compete for shared representations, reducing performance in some domains (Yu et al., 2020). Moreover, training GFMs across tasks is computationally expensive, requiring efficient parameter sharing and scalable architectures to manage resource demands (Huang et al., 2023).

**Mixture of Experts GFMs.** The Mixture of Experts (MoE) paradigm (bin Cai et al., 2024) leverages the heterogeneity of graph datasets by dynamically routing each to a suitable expert network, effectively classifying and assigning datasets to specialized solutions. AnyGraph (Xia & Huang, 2024) applies a linear MoE with singular value decomposition for inductive link prediction, using structural and feature-based routing scores to select the best expert. GraphAny (Zhao et al., 2025) focuses on node classification, where each expert is a non-parametric graph representation solved analytically via least squares, and combined using an attention mechanism trained across all expert outputs. While these methods excel in dataset-specific adaptation, their applicability remains limited to single-task settings. Extending MoE frameworks to support multiple graph tasks remains an open challenge.

# 3 EveryGraph: Random Projection as a Unifying Step Towards Graph Foundation Models

In this section, we introduce EveryGraph, designed specifically to address the dual challenges of GFMs discussed in Section 1. The overall architecture is illustrated in Figure 3. In high-level terms, the data flows in EveryGraph as follows: An input graph $G = \left(A, X \in \mathbb{R}^{n \times d}\right)$ is first embedded in a theoretically-grounded unified representation space using a random projection (Subsection 3.1), yielding $\hat{G} = \left(A, \hat{X} \in \mathbb{R}^{n \times k}\right)$. We then utilize a Graph-UNet (Gao & Ji, 2019), pretrained on link prediction (Subsection 3.2), to build a representation $\tilde{G} = \left(A, \tilde{X} \in \mathbb{R}^{n \times l}\right)$. Finally, inference on the downstream task is done in an in-context manner using closed-form solutions (Subsection 3.3).

As our experiments in Section 4 will later show, these components enable a single model to process diverse graph structures representing different domains, laying the groundwork for a full-fledged foundation model for graphs.

## 3.1 Random Projection for Unified Representation Space

Graph datasets vary widely in feature dimensions and meaning. Neural networks, however, are structurally fixed: once configured and trained, they cannot easily adapt to changes in input format.

To bridge this gap, EveryGraph first embeds the input features in unified representation space of dimension $k$ using a random projection followed by $L_2$ normalization. Formally, node features $x_i \in \mathbb{R}^d$ are transformed to

$$\hat{x}_i = \frac{A x_i}{\|A x_i\|_2} \in \mathbb{R}^k \tag{1}$$

Random Gaussian projection are well-studied operators, which provide an important theoretic property: They preserve pairwise similarity between data points. In Appendix A, we formally prove that the described input features transformation preserves the cosine similarity between two graph nodes. For ease of presentation, we ~~summaries~~ summarize the conclusion in Corollary 1.

**Corollary 1** (EveryGraph's Unified Representation Space Preserves Cosine Similarity)**.** *Let $x, y \in \mathbb{R}^d$ and let $A \in \mathbb{R}^{k \times d}$ be a random matrix with entries $A_{ij} \sim \mathcal{N}(0,1)$. Denote the $L_2$ normalized projections $\hat{x} = \frac{Ax}{\|Ax\|}$, $\hat{y} = \frac{Ay}{\|Ay\|}$. Then for sufficiently large $k$,*

$$\|\hat{x} - \hat{y}\|_2 \approx 2(1 - CosineSimilarity(x, y)) \tag{2}$$

*with high probability.*

It follows that the unified representation space constructed by the transformation described in Eq. 1 preserves and encodes the similarity between nodes in the original input graph. Proof can be found in the appendix A

**Similarity to the Johnson-Lindenstrauss lemma.** It may be observed that Corollary 1, together with the theorem from which it is derived, closely resembles the classical Johnson–Lindenstrauss (JL) lemma, which provides analogous guarantees for a related class of transformations.

**Other Approaches for Unified Representation Space.** As discussed, in order to allow a single model to process datasets with differing feature dimensions, it is essential to pre-process the input features in a way that preserves core information while standardizing the nodes feature size. Another approach to achieve this goal is to use PCA with $k$ dimension. In Appendix D, we present an ablation study comparing both methods, which demonstrates the superiority of random projection. Another drawback of this approach is that it is ill-defined for datasets with feature dimension $d < k$.

## 3.2 Link Prediction Self-Supervised Pre-Training

We capitalize on the random projection's proven benefits by training a Graph-UNet (Gao & Ji, 2019) network on link prediction task, to which node similarity is intrinsic.

Link prediction was chosen as the pretraining task over other self-supervised methods because it forces the model to learn and preserve the underlying relational structure (Zhou et al., 2020). This property is critical for a GFM, which has to generalize across datasets and domains by building a universal representation of connectivity patterns, independent of node semantics. Since the topology (i.e., the existence of an edge between two given nodes) is given by definition for every graph dataset, it allows EveryGraph to train on every such dataset, regardless of its designated task.

It also follows that our framework allows flexibility in selecting the dataset used to train the embedding model.

We chose Graph-UNet for its ability to preserve graph structure, essential to downstream tasks (see Appendix B for more in-depth discussion). An illustration of our training process is shown in Figure 2.

In inference time, this module will be used to perform link-prediction tasks. Other tasks will utilize the node representation it generates as input, which we denote

$$\tilde{X} = \text{GraphUNet}_\theta\left(A, \hat{X}\right) \tag{3}$$

### 3.3 Closed-Form In-Context Downstream Adaptation

Inspired by language modeling, EveryGraph enables in-context generalization to unseen datasets and tasks by leveraging closed-form inference techniques. For a given dataset, the node representations for training split generated by EveryGraph, $\tilde{X}_{\text{train}}$, are used to compute those close-form solutions, adapting the model. Figure 3 illustrates EveryGraph's inference process.

**Message Passing: Final Embeddings.** The output node embeddings obtained from the link prediction model undergo a final non-parametric embedding process. In this process, we obtain the message-passed features matrix

$$\check{X} = \left(D^{-1}A\right)^2 \tilde{X} \in \mathbb{R}^{n \times l} \tag{4}$$

where $D$ is the degree matrix and $A$ is the adjacency matrix. We emphasize that this processing stage is taken for all nodes, regardless of train-test split.

**Ridge Regression Solution.** EveryGraph utilizes a ridge regression closed-form solution for node and graph classification problems. A standard ridge regression problem is defined as:

$$\hat{\beta} = \underset{\beta}{\text{argmin}}\left(\|\mathbf{Y} - \mathbf{X}\beta\|_2^2 + \lambda\|\beta\|_2^2\right) \tag{5}$$

where $\mathbf{Y} \in \mathbb{R}^n$ is the vector of target values, $\mathbf{X} \in \mathbb{R}^{n \times l}$ is the features matrix containing the feature values, $\beta \in \mathbb{R}^l$

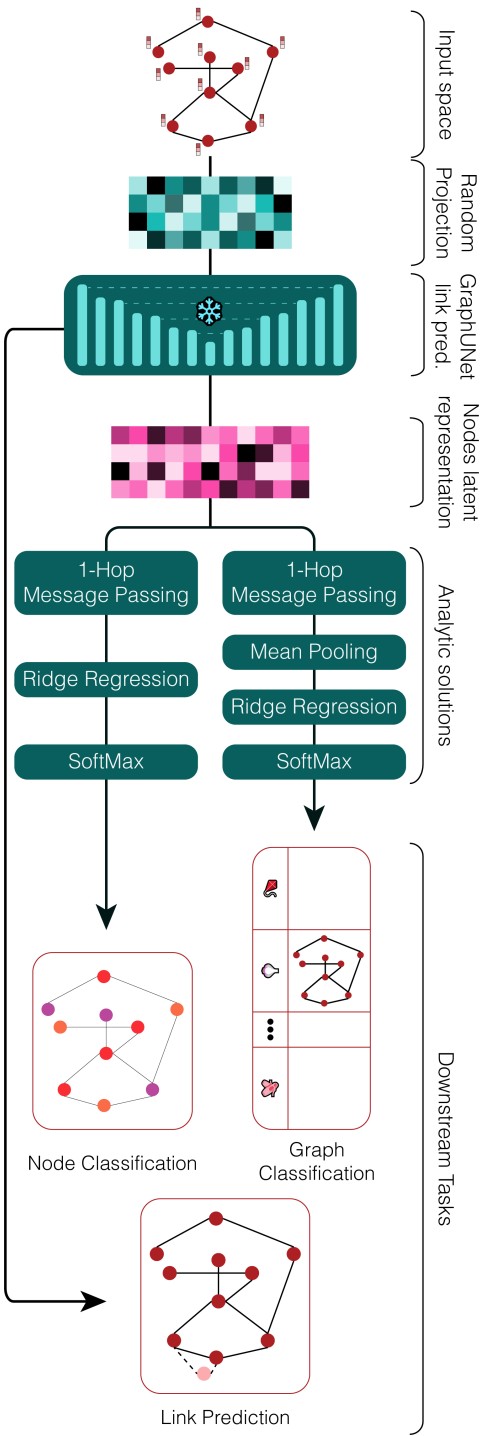

Figure 3: EveryGraph inference. Graph data is randomly projected into a unified space, encoded by a GraphUNET trained for link prediction, and processed via closed-form solutions for final task predictions.

Table 2: Link prediction performance. Best in-context method on each dataset is marked in **bold**.

| Category | Dataset →
Method ↓ | SQUIRREL
ROC-AUC ↑ | CHAMELEON
ROC-AUC ↑ | WIKI
ROC-AUC ↑ | AIREU
ROC-AUC ↑ | AIRUS
ROC-AUC ↑ |
|---|---|---|---|---|---|---|
| Supervised | MLP (Huang et al., 2021) | $0.81_{\pm 0.00}$ | $0.91_{\pm 0.01}$ | $0.88_{\pm 0.00}$ | $0.70_{\pm 0.12}$ | $0.84_{\pm 0.01}$ |
| | GCN (Kipf & Welling, 2017) | $0.71_{\pm 0.13}$ | $0.89_{\pm 0.00}$ | $0.82_{\pm 0.03}$ | $0.64_{\pm 0.02}$ | $0.85_{\pm 0.01}$ |
| | GAT (Veličković et al., 2018) | $0.79_{\pm 0.02}$ | $0.86_{\pm 0.00}$ | $0.85_{\pm 0.01}$ | $0.56_{\pm 0.02}$ | $0.79_{\pm 0.01}$ |
| | GATv2 (Brody et al., 2022) | $0.81_{\pm 0.017}$ | $0.88_{\pm 0.02}$ | $0.87_{\pm 0.02}$ | $0.54_{\pm 0.02}$ | $0.78_{\pm 0.01}$ |
| | GIN (Xu et al., 2019) | $0.81_{\pm 0.05}$ | $0.94_{\pm 0.00}$ | $0.87_{\pm 0.01}$ | $0.57_{\pm 0.02}$ | $0.77_{\pm 0.02}$ |
| In-Context | EveryGraph (Cora) | $\mathbf{0.82_{\pm 0.05}}$ | $\mathbf{0.86_{\pm 0.06}}$ | $0.93_{\pm 0.02}$ | $0.75_{\pm 0.12}$ | $\mathbf{0.75_{\pm 0.12}}$ |
| | EveryGraph (Pubmed) | $0.81_{\pm 0.01}$ | $0.86_{\pm 0.04}$ | $\mathbf{0.93_{\pm 0.01}}$ | $\mathbf{0.78_{\pm 0.03}}$ | $0.69_{\pm 0.09}$ |
| | EveryGraph (CoPhysics) | $0.79_{\pm 0.02}$ | $0.83_{\pm 0.06}$ | $0.92_{\pm 0.01}$ | $0.75_{\pm 0.02}$ | $0.65_{\pm 0.05}$ |

is the vector of regression coefficients, and $\lambda$ is the regularization parameter, controlling the strength of the penalty on the coefficients. The first term $\|\mathbf{Y} - \mathbf{X}\beta\|_2^2$ represents the residual sum of squares (RSS), while the second term $\lambda\|\beta\|_2^2$ is the $L_2$ regularization coefficient. This problem has a well-known closed-form solution:

$$\hat{\beta} = (\mathbf{X}^T\mathbf{X} + \lambda I)^{-1}\mathbf{X}^T\mathbf{Y} \tag{6}$$

Note that for this can be adapted to classification tasks by setting $\mathbf{Y} \in \mathbb{R}^{n \times c}$ to be a one-hot encoding of the target class, $c$ being the number of classes. If follows that the coefficients are now a matrix of shape $l \times c$.

**Node Classification.** In datasets designed for node level tasks (such as node classification), the input graph is split to train, validation, and test nodes. To adapt EveryGraph to a node-level dataset, we use the training nodes to construct $\mathbf{X} = \check{X}_{\text{train}}, \mathbf{Y} = Y_{\text{train}}$ that to find the optimal $\hat{\beta}$ per Equation 6. After finding $\hat{\beta}$, EveryGraph is fully adapted to the new task, and the final logits can be formally written as

$$\hat{Y}_{\text{test}} = \check{X}_{\text{test}}\hat{\beta} \tag{7}$$

**Graph-Level Tasks.** Graph-level tasks were shown to benefit significantly from expressive graph-level embeddings (Xu et al., 2019; Ying et al., 2018). In EveryGraph, we incorporate several critical features into our graph representations. First, we include Random Walk Positional Encodings (RWPE) for each node, as introduced in (Dwivedi & Bresson, 2021). Second, we leverage geometric features of the graph by counting cycles: each node is annotated with the number of 3-, 4-, and 5-cycles it participates in. Additionally, we compute global counts of 3-, 4-, 5-, and 6-cycles and concatenate them to all node features in the graph, enriching each node's feature space with structural context. RWPE, local and global cycle counts are concatenated to the node representation prior to the message passing operation. Given the embedding of these statistics $X_{\text{RWPE}} \in \mathbb{R}^{n \times d_1}$, $X_{\text{cycles-local}} \in \mathbb{R}^{n \times d_2}$, $X_{\text{cycles-global}} \in \mathbb{R}^{n \times d_3}$,

$$\tilde{X}^{\text{graph}} = \tilde{X}\|X_{\text{RWPE}}\|X_{\text{cycles-local}}\|X_{\text{cycles-global}} \in \mathbb{R}^{n \times l'} \tag{8}$$

Where $\|$ denotes the concatenation operator and $l' = l + d_1 + d_2 + d_3$

Unlike node classification, in graph classification tasks labels are assigned entire graphs rather than individual nodes. We obtain the graph embedding via mean pooling across the nodes. The final embedding of a set of graphs is

$$\check{X} = \text{Concat}\left(..., \text{mean}\left(\left(D_i^{-1}A_i\right)^2 \tilde{X}_i^{\text{graph}}\right), ...\right) \in \mathbb{R}^{g \times k} \tag{9}$$

where $g$ is the number of graphs in the dataset (or batch, where applicable). This enables us to assign a single vector for each graph in the dataset and use the ridge regression solution in a similar way to the node classification task.

## 4 Experiments

We evaluate EveryGraph on more than 20 datasets across link prediction, node classification, and graph classification, and present the results in this section. Datasets and their statistics are given in Appendix E.

Table 3: Performance on node-level tasks. Best *in-context method* on each dataset is marked in **bold**.

| Category | Dataset → 
 Method ↓ | ARXIV 
 ACCURACY ↑ | CORA 
 ACCURACY ↑ | PUBMED 
 ACCURACY ↑ | WIKI 
 ACCURACY ↑ | COPHYSICS 
 ACCURACY ↑ |
|---|---|---|---|---|---|---|
| Supervised | MLP (HUANG ET AL., 2021) | $50.77_{\pm0.125}$ | $55.23_{\pm00.42}$ | $72.83_{\pm0.12}$ | $61.53_{\pm1.29}$ | $94.15_{\pm0.13}$ |
| | GCN (Kipf & Welling, 2017) | $68.47_{\pm0.26}$ | $81.73_{\pm0.39}$ | $78.53_{\pm0.17}$ | $64.60_{\pm0.39}$ | $95.39_{\pm0.02}$ |
| | GAT (Veličković et al., 2018) | $70.16_{\pm0.13}$ | $81.73_{\pm0.39}$ | $77.93_{\pm0.23}$ | $63.52_{\pm1.90}$ | $94.31_{\pm0.28}$ |
| | GATv2 (Brody et al., 2022) | $70.13_{\pm0.21}$ | $81.73_{\pm0.63}$ | $77.96_{\pm0.39}$ | $63.47_{\pm2.82}$ | $95.27_{\pm0.51}$ |
| | GIN (Xu et al., 2019) | $70.29_{\pm0.21}$ | $80.18_{\pm1.21}$ | $76.80_{\pm0.22}$ | $60.02_{\pm3.74}$ | $93.53_{\pm0.72}$ |
| | Graphormer (Ying et al., 2021) | $71.91_{\pm0.23}$ | $82.18_{\pm0.43}$ | $78.29_{\pm0.74}$ | $63.15_{\pm1.43}$ | $95.74_{\pm0.13}$ |
| In-Context | GraphAny (Cora) (Zhao et al., 2025) | $58.59_{\pm0.09}$ | $80.13_{\pm0.06}$ | $76.5_{\pm0.2}$ | $56.05_{\pm2.46}$ | $92.61_{\pm0.51}$ |
| | GraphAny (Pubmed) (Zhao et al., 2025) | $58.54_{\pm0.07}$ | $80.09_{\pm0.07}$ | $77.02_{\pm0.08}$ | $57.79_{\pm0.23}$ | $92.58_{\pm0.32}$ |
| | EveryGraph (Cora) | $64.39_{\pm0.14}$ | $\mathbf{80.42_{\pm0.34}}$ | $76.68_{\pm0.88}$ | $60.68_{\pm0.97}$ | $94.54_{\pm0.06}$ |
| | EveryGraph (CoPhysics) | $64.33_{\pm0.20}$ | $79.02_{\pm0.53}$ | $76.68_{\pm1.25}$ | $60.52_{\pm1.36}$ | $\mathbf{94.56_{\pm0.05}}$ |
| | EveryGraph (Pubmed) | $\mathbf{64.44_{\pm0.17}}$ | $79.22_{\pm0.73}$ | $\mathbf{77.49_{\pm0.73}}$ | $\mathbf{61.89_{\pm1.37}}$ | $94.52_{\pm0.08}$ |

In our experiments, we seek answers to the following questions:

- How does our proposed method, using in-context adaptation, perform compared to supervised, established baselines across various tasks?

- Does our method generalize effectively across graphs of different sizes, sparsity levels, and task-specific datasets?

- As suggested by Corollary 1, is the information required for meaningful learning preserved under the random projection paradigm?

- How does EveryGraph handle link prediction on heterophilous benchmarks?

- Is EveryGraph's performance robust to the choice of trainable backbone, or is it tied to the Graph U-Net specifically?

- How sensitive is EveryGraph to the dimension $k$ of the random projection that unifies heterogeneous feature spaces?

**Experimental Setup.** We evaluate our method against several widely used fully-supervised baselines, including MLP, GCN, GAT, GATv2, GIN, and Graphormer, as well as GraphAny (Zhao et al., 2025), a recent in-context approach. All experiments are conducted using the GraphSaintNodeSampler (Zeng et al., 2020), with a consistent batch size of 512 nodes and an 70-10-20 train-val-test split for link prediction. For node classification we either maintain the given split or create our own split for datasets that lack one. In our node classification splits we maintain a 5% training split for the ridge regression. For graph classification, we apply an 80/20 train–test split across all methods and evaluate performance using 10-fold cross-validation. The evaluations are performed on a single NVIDIA RTX3090 GPU, in Appendix C we elaborate on the computational complexity of EveryGraph- specifically on for the closed ridge regression solution. We assess the performance of EveryGraph on link prediction, node classification, and graph classification, using a diverse set of datasets spanning citation networks, social networks, neuroscience networks, e-commerce graphs, and more. Additional details on training procedures and implementation specifics are provided in Appendix G, and the rationale behind each architectural component choice is elaborated in the coming subsections and Appendix D.

## 4.1 Node Classification Results

Table 3 presents the in-context performance of EveryGraph across multiple datasets. In each experiment, the model is trained on the dataset listed in parentheses for link prediction and evaluated, without any fine-tuning, on all other datasets shown in the table. A comprehensive set of results for all datasets is available in Appendix D. Each classical method (e.g., MLP, GCN, etc.) is trained separately, using hyperparameters and training configurations which is disclosed in the appendix F. The results for GraphAny (Zhao et al.,

Table 4: Graph classification accuracy. Best in-context method on each dataset is marked in **bold**.

| Category | Dataset → Method ↓ | MUTAG Acc. ↑ | PROTEINS Acc. ↑ | NCI1 Acc. ↑ | NCI109 Acc. ↑ |
|---|---|---|---|---|---|
| Supervised | MLP (Huang et al., 2021) | $75.44_{\pm4.47}$ | $72.05_{\pm0.76}$ | $66.02_{\pm0.51}$ | $65.01_{\pm0.94}$ |
| | GCN (Kipf & Welling, 2017) | $76.32_{\pm7.75}$ | $71.15_{\pm1.84}$ | $67.19_{\pm1.90}$ | $67.68_{\pm1.92}$ |
| | GAT (Veličković et al., 2018) | $75.44_{\pm5.41}$ | $68.16_{\pm5.53}$ | $66.46_{\pm1.44}$ | $60.13_{\pm4.84}$ |
| | GATv2 (Brody et al., 2022) | $78.07_{\pm4.96}$ | $70.40_{\pm2.40}$ | $61.64_{\pm9.14}$ | $64.89_{\pm5.37}$ |
| | GIN (Xu et al., 2019) | $88.60_{\pm1.23}$ | $73.69_{\pm1.65}$ | $77.58_{\pm0.41}$ | $76.80_{\pm0.98}$ |
| | Graphormer (Ying et al., 2021) | $75.44_{\pm2.48}$ | $58.45_{\pm13.73}$ | $59.69_{\pm1.49}$ | $57.43_{\pm4.12}$ |
| In-Context | GraphAny (Cora) (Zhao et al., 2025) | $66.66_{\pm4.44}$ | $59.67_{\pm6.19}$ | $50.49_{\pm2.28}$ | $50.85_{\pm1.26}$ |
| | GraphAny (Pubmed) (Zhao et al., 2025) | $58.12_{\pm2.42}$ | $60.12_{\pm2.83}$ | $49.15_{\pm0.40}$ | $51.53_{\pm0.40}$ |
| | EveryGraph (Cora) | $75.26_{\pm4.58}$ | $61.34_{\pm3.56}$ | $59.46_{\pm1.70}$ | $58.11_{\pm1.16}$ |
| | EveryGraph (CoPhysics) | $74.21_{\pm5.10}$ | $\mathbf{61.43_{\pm3.85}}$ | $59.65_{\pm1.33}$ | $58.18_{\pm1.01}$ |
| | EveryGraph (Pubmed) | $\mathbf{75.26_{\pm4.27}}$ | $60.71_{\pm3.45}$ | $\mathbf{59.80_{\pm1.52}}$ | $\mathbf{58.45_{\pm0.93}}$ |

2025) were generated by us using the same evaluation protocol as EveryGraph, with the model trained on the dataset specified in parentheses. We note that GraphAny is a node-classification only framework, trained and designed for it exclusively.

The results in Table 3 show that EveryGraph achieves competitive, and in some cases, superior performance compared to both in-context and fully supervised baselines. This supports the assertion made in Corollary 1 and Table 1, demonstrating that the random projection approach preserves critical information. Importantly, this finding holds across a diverse range of domains, including citation networks, e-commerce graphs, and traffic networks.

In Appendix F, we assess the results of task-and-dataset-specific fine tuning, where improved results are achieved with a computational cost that is in conflict with our main objective.

### 4.2 Link Prediction

As previously mentioned, EveryGraph adopts link prediction as its training objective. This task follows a contrastive learning paradigm, where similar node embeddings are drawn closer together while dissimilar ones are pushed apart. We evaluate our link prediction performance using the ROC-AUC metric, as shown in Table 2. We decided to focus this task on heterophilous graphs specifically to test our framework's expressive power against such datasets. The results indicate that EveryGraph achieves competitive performance compared to classical methods, and in some cases superior (Squirrel and WIKI), even though it is trained on a single dataset. Additionally, the findings support the claim from Corollary 1 that the random projection approach preserves essential information. Table 6 shows the effects of the pretraining dataset, over the EveryGraph's results. As we can see while node and graph level tasks are less effected by the pretraining dataset, link prediction is. We attribute this to the fact that we managed to provide a feature level unified space which has more to do with node and graph level tasks, where link is effected more by the structural habit of graphs.

### 4.3 Graph Classification

As with the previously discussed tasks, EveryGraph is evaluated 4 in a *in-context* setting: it is trained on one dataset and tested on others without any fine-tuning. The same evaluation protocol is applied to the GraphAny results shown in the table. In contrast, all baseline methods are trained individually on each dataset, and their best-performing results are reported. While the results show that EveryGraph can generalize to the graph classification task, its performance lags behind that of classical methods on some datasets.

This gap, however, conflates two effects: the graph-level adaptation mechanism itself and a severe distribution shift, since the setting above transfers a model pretrained on a citation network to small molecular graphs

that differ sharply in size, topology, and feature semantics. To disentangle them, we repeat the evaluation with in-domain pretraining, and find that a substantial part of the gap is attributable to distribution shift rather than to the adaptation mechanism. Concretely, we pretrain EveryGraph *within* the molecular domain, on MUTAG, and transfer to held-out molecular datasets, comparing against the citation-pretrained (Cora) variant under the identical evaluation protocol.

Table 5: In-domain vs. cross-domain pretraining for graph classification (accuracy). The MUTAG-pretrained model transfers within the molecular domain; the Cora-pretrained model transfers from a citation network.

| Pretraining dataset | MUTAG | PROTEINS | NCI1 | NCI109 |
|---|---|---|---|---|
| EveryGraph (Cora) | $75.26_{\pm 4.58}$ | $61.34_{\pm 3.56}$ | $59.46_{\pm 1.70}$ | $58.11_{\pm 1.16}$ |
| EveryGraph (MUTAG) | $\mathbf{77.30_{\pm 6.16}}$ | $\mathbf{62.89_{\pm 1.60}}$ | $\mathbf{64.87_{\pm 3.10}}$ | $\mathbf{65.65_{\pm 2.93}}$ |

As shown in Table 5, in-domain pretraining improves performance across all four molecular targets (for example, NCI109 rises from 58.11 to 65.65). This indicates that a substantial part of the graph-classification gap reported in the main text is attributable to the citation-to-molecule distribution shift rather than to a limitation of the graph-level adaptation mechanism. Accordingly, we qualify our cross-domain graph-classification claim: EveryGraph adapts effectively to graph-level tasks when the pretraining and target domains are compatible, while cross-domain transfer across drastically different graph distributions remains an open direction for future work.

## 4.4 Assessing Information Preservation

To definitively examine the information preservation capability of the random projection technique, we conducted a new set of ablation experiments. We trained three different backbone GNN architectures (GCN, GAT, and GATv2) on several datasets under the following three scenarios:

1. Original Features (Original): Training the model using the original node features (**Features**).

2. Random Projection (RandProj): Training the model using the features generated by our random projection paradigm (**RandProj = Features $\times$ N**). Where N is a matrix whose values are taken from the normal distribution.

3. Random Noise Features (RNF): Training the model using randomly generated node features (**RNF**), completely replacing the original features.

The results, presented in Table 1, show that models trained using our random projection features (**RandProj**) maintain performance remarkably close to models trained on the original features (**Features**) and in some cases even precedes it. This stands in stark contrast to the significantly degraded performance seen when using pure random features (**RNF**). This experimental evidence strongly validates our theoretical claim that key graph structure and feature information is preserved under the random projection paradigm, thereby establishing a causal connection between the projection technique and EveryGraph's superior generalization ability.

## 4.5 Effect of the Backbone Architecture

EveryGraph uses a Graph U-Net (Gao & Ji, 2019) as its trainable backbone (Section 3.2). To assess how much of the reported performance stems from the EveryGraph framework itself, as opposed to this particular backbone choice, we replace the Graph U-Net with two standard message-passing architectures, GCN (Kipf & Welling, 2017) and GraphSAGE (Hamilton et al., 2017), while keeping the rest of the pipeline (random projection, link-prediction pretraining, and closed-form adaptation) fixed. Each backbone is pretrained on link prediction and evaluated in the same in-context manner.

Table 7 shows that the framework operates with standard message-passing backbones: GCN in particular is close to the Graph U-Net on node classification, while GraphSAGE lags on link prediction. The Graph U-Net

Table 6: Task performance average vs the pretraining dataset. Best task results regarding the pretraining graph are in bold

| Task→ PreTrainDataset ↓ | LINK ROC-AUC ↑ | NODE ACCURACY ↑ | GRAPH ACCURACY ↑ |
|---|---|---|---|
| Pubmed | $0.85_{\pm 0.01}$ | $62.85_{\pm 0.21}$ | $\mathbf{65.92_{\pm 2.45}}$ |
| AmzComp | $0.66_{\pm 0.04}$ | $\mathbf{63.07_{\pm 0.32}}$ | $64.45_{\pm 1.18}$ |
| FCora | $\mathbf{0.85_{\pm 0.01}}$ | $62.88_{\pm 0.21}$ | $65.50_{\pm 3.08}$ |

Table 7: Backbone ablation. The EveryGraph pipeline is held fixed while the trainable backbone is varied. Link prediction is reported as ROC-AUC and node classification as accuracy.

| Test Dataset | Link Prediction (ROC-AUC) ↑ | | | Node Classification - Accuracy ↑ | | |
|---|---|---|---|---|---|---|
| | GCN | GraphSAGE | Graph U-Net (ours) | GCN | GraphSAGE | Graph U-Net (ours) |
| Arxiv | $84.71_{\pm 0.58}$ | $70.85_{\pm 4.77}$ | $\mathbf{85.91_{\pm 0.82}}$ | $\mathbf{64.46_{\pm 0.11}}$ | $62.62_{\pm 0.18}$ | $\mathbf{64.46_{\pm 0.09}}$ |
| CoPhysics | $94.40_{\pm 0.19}$ | $69.67_{\pm 1.04}$ | $\mathbf{95.13_{\pm 1.33}}$ | $94.52_{\pm 0.08}$ | $93.94_{\pm 0.07}$ | $\mathbf{94.57_{\pm 0.08}}$ |
| WkCS | $80.27_{\pm 0.54}$ | $79.14_{\pm 1.16}$ | $\mathbf{80.76_{\pm 0.32}}$ | $73.24_{\pm 0.33}$ | $68.40_{\pm 0.23}$ | $\mathbf{73.37_{\pm 0.39}}$ |

is consistently best, but by a modest and stable margin, which we attribute to its hierarchical pooling and multi-resolution representations (Appendix B). The reported results therefore follow from the EveryGraph framework rather than solely from the backbone. We further note that our comparison to GraphAny (Zhao et al., 2025) uses different convolutional operators; the present ablation clarifies that our framework is not tied to a single backbone, and that this architectural difference is unlikely to account for the observed advantage.

## 4.6 Sensitivity to the Projection Dimension $k$

The random projection maps input features of arbitrary dimension $d$ into a fixed dimension $k$ (Section 3.1). Since our datasets have heterogeneous feature dimensions, we study the sensitivity of EveryGraph to $k$. We emphasize that $k = 1024$ was fixed *a priori* for all experiments above; no target-dataset information was used to select it. Here we sweep $k \in \{8, 16, 32, 64, 128, 256, 512, 1024, 2048\}$ on datasets of differing original feature dimension.

Table 8: Link prediction (ROC-AUC) as a function of the projection dimension $k$.

| Dataset | $k=8$ | $k=16$ | $k=32$ | $k=64$ | $k=128$ | $k=256$ | $k=512$ | $k=1024$ | $k=2048$ |
|---|---|---|---|---|---|---|---|---|---|
| CoCS | 0.760 | 0.803 | 0.846 | 0.865 | 0.873 | 0.920 | 0.917 | **0.939** | 0.933 |
| Citeseer | 0.740 | 0.781 | 0.804 | 0.833 | 0.855 | 0.890 | 0.899 | **0.917** | 0.909 |
| Pubmed | 0.762 | 0.815 | 0.854 | 0.865 | 0.878 | 0.918 | 0.921 | **0.934** | 0.931 |

Tables 8 and 9 show a consistent trend across both tasks and all datasets: performance increases with $k$ and saturates around $k = 1024$–$2048$, with only minor fluctuations near the plateau. This matches the theory: Theorem 1 requires $k = O(\ln N/\varepsilon^2)$ for the similarity-preservation guarantee, so larger $k$ tightens the guarantee with diminishing returns once the representation space is sufficiently large. Because performance is stable and near-optimal for large $k$, increasing $k$ is uniformly safe, which motivates our fixed, data-independent choice of $k = 1024$ rather than any per-dataset tuning.

## 5 Conclusion

In this work, we introduced EveryGraph, a novel graph neural architecture aimed at addressing the long-standing generalization challenge in graph learning. By leveraging a theoretically grounded unified repre-

Table 9: Node classification (accuracy) as a function of the projection dimension $k$.

| Dataset | $k=8$ | $k=16$ | $k=32$ | $k=64$ | $k=128$ | $k=256$ | $k=512$ | $k=1024$ | $k=2048$ |
|---------|-------|--------|--------|--------|---------|---------|---------|----------|----------|
| FCora | 20.74 | 32.45 | 40.16 | 45.78 | 50.31 | 52.96 | 54.24 | **54.57** | 54.28 |
| Citeseer | 37.45 | 45.91 | 51.55 | 56.60 | 60.92 | 63.25 | 65.07 | 67.55 | **67.92** |
| CoCS | 63.33 | 77.19 | 83.66 | 87.36 | 89.06 | 89.90 | **90.18** | 90.08 | 89.90 |

sentation space, EveryGraph enables effective in-context inference across a wide range of datasets and tasks. Over this space, we employed a GraphUNet, trained on a link prediction task, with closed-form solutions using the latent embeddings for additional graph tasks. Crucially, EveryGraph achieves this while operating with a training budget that is $10\times$ to $20\times$ lower than that required by many fully supervised approaches.

**Limitations**  While EveryGraph shows strong generalization with minimal overhead, we identify key limitations. On graph classification it currently trails specialized methods under large cross-domain shifts, although our analysis (Section 4) indicates that much of this gap stems from the distribution shift between pretraining and target domains rather than from the graph-level adaptation itself, and narrows substantially with in-domain pretraining. Its expressive power is also restricted by ridge regression, potentially constraining the capture of complex relational patterns.

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

## A    Theoretic Foundations of the Unified Representation Space

In this section, we prove that the unified representation space as constructed in Subsection 3.1 preserves the cosine similarity between embedded vectors.

**Notation.**    We denote $\text{CosineSimilarity}(x, y) = \frac{x^T y}{\|x\| \|y\|}$.

**Lemma 1** (Distribution of Projected Coordinates). *Let $x, y \in \mathbb{R}^d$ be unit vectors. Let $A \in \mathbb{R}^{k \times d}$ be a random matrix with entries $A_{ij} \sim \mathcal{N}(0, 1)$. Let $u = Ax$ and $v = Ay$ be the unnormalized projected vectors. Then, for any row index $i \in \{1, \ldots, k\}$, the pair of coordinates $(u_i, v_i)$ follows a bivariate normal distribution:*

$$\begin{pmatrix} u_i \\ v_i \end{pmatrix} \sim \mathcal{N} \left( \begin{pmatrix} 0 \\ 0 \end{pmatrix}, \begin{pmatrix} 1 & \text{CosineSimilarity}(x, y) \\ \text{CosineSimilarity}(x, y) & 1 \end{pmatrix} \right) \tag{10}$$

*Furthermore, the pairs $\{(u_i, v_i)\}_{i=1}^k$ are independent and identically distributed (i.i.d.).*

*Proof.* The component $u_i = \sum_{j=1}^d A_{ij} x_j$ is a linear combination of independent Gaussian variables, and thus is Gaussian. Following the same argument, $v_i$ is Gaussian. Since $A_{ij}$ are centered at 0, the expectations are $E[u_i] = E[v_i] = 0$. The variance of $u_i$ (and similarly, of $v_i$) is

$$\text{Var}(u_i) = E\left[u_i^2\right] - \underbrace{E\left[u_i^2\right]}_{= \, 0} = \|x\|_2^2 E[A_{ij}^2] = 1 \cdot 1 = 1$$

The covariance is

$$\text{Cov}\,(u_i, v_i) = E\left[\left(u_i - \underbrace{E[u_i]}_{= \, 0}\right)\left(v_i - \underbrace{E[v_i]}_{= \, 0}\right)\right]$$

$$= E[u_i v_i] = x^T E[A_i^T A_i] y = x^T I y = \text{CosineSimilarity}(x, y)$$

The last equality follows from $x, y$ being unit vectors.
The independence across $i$ follows directly from the independence of the rows of the random matrix $A$. $\quad\square$

**Lemma 2** (Concentration of Terms). *Let $x, y \in \mathbb{R}^d$ be unit vectors. Let $A \in \mathbb{R}^{k \times d}$ be a random matrix with entries $A_{ij} \sim \mathcal{N}(0, 1)$. Let $u = Ax$ and $v = Ay$ be the unnormalized projected vectors. Define the sample averages $S_{uu} = \frac{1}{k}\|u\|_2^2$, $S_{vv} = \frac{1}{k}\|v\|_2^2$, and $S_{uv} = \frac{1}{k} u^T v$. For brevity, denote $\rho = \text{CosineSimilarity}(x, y)$.*

*For any $\varepsilon \in (0, 1)$, there exists a constant $C$ such that the following concentration bounds hold:*

$$\Pr(|S_{uu} - 1| \geq \varepsilon) \leq 2 \exp\left(-\frac{k\varepsilon^2}{8}\right) \tag{11}$$

$$\Pr(|S_{vv} - 1| \geq \varepsilon) \leq 2 \exp\left(-\frac{k\varepsilon^2}{8}\right) \tag{12}$$

$$\Pr(|S_{uv} - \rho| \geq \varepsilon) \leq 2 \exp\left(-\frac{k\varepsilon^2}{C}\right) \tag{13}$$

*Proof. Concentration of Self-Terms ($S_{uu}, S_{vv}$):* Recall that $u_i \sim \mathcal{N}(0, 1)$ i.i.d. Thus, the sum $kS_{uu} = \sum_{i=1}^k u_i^2$ follows a Chi-squared distribution with $k$ degrees of freedom, denoted $\chi_k^2$. Apply the Laurent-Massart concentration bound (Laurent & Massart, 2000) for $\chi_k^2$ variables. For any $\varepsilon \in (0, 1)$:

$$\Pr\left(\left|\frac{\chi_k^2}{k} - 1\right| \geq \varepsilon\right) \leq 2 \exp\left(-\frac{k\varepsilon^2}{8}\right)$$

This proves inequalities (11) and (12).

*Concentration of Cross-Term ($S_{uv}$):* Consider the term $S_{uv} = \frac{1}{k}\sum_{i=1}^{k} Z_i$, where $Z_i = u_i v_i$. The variables $Z_i$ are independent with mean $E[Z_i] = \rho$. Recall that this is a cosine similarity term, thus $\rho^2 \leq 1$. Since $u_i, v_i$ are standard Gaussians, their product $Z_i$ is a *sub-exponential* random variable, with variance

$$\sigma^2 \triangleq \text{Var}(Z_i) = E[u_i^2 v_i^2] - (E[u_i v_i])^2 = (1 + 2\rho^2) - \rho^2 = 1 + \rho^2 \leq 2 \tag{14}$$

Apply Bernstein's inequality for some absolute constant $K$:

$$\Pr\left(\left|\frac{1}{k}\sum Z_i - \rho\right| \geq \varepsilon\right) \leq 2\exp\left(-\frac{k\varepsilon^2}{2\sigma^2 + 2K\varepsilon}\right) \tag{15}$$

Replacing $\sigma^2$ with its upper bound (Eq. 14) and considering small $\varepsilon$ (where the $\varepsilon^2$ term in the exponent dominates), we can absorb the variance and the sub-exponential parameter $K$ into a single universal constant $C$. Thus, we obtain a bound dependent only on $k$ and $\varepsilon$, yielding the desired

$$\Pr\left(|S_{uv} - \rho| \geq \varepsilon\right) \leq 2\exp\left(-\frac{k\varepsilon^2}{C}\right) \tag{16}$$

$\square$

**Lemma 3** (Concentration Bound for Squared Distance). *Let $\hat{x} = \frac{u}{\|u\|_2}$ and $\hat{y} = \frac{v}{\|v\|_2}$, $u, v$ follow their definition in Lemma 2. For any pair $x, y$ and error tolerance $\varepsilon \in (0, 1)$, there exists a constant $C'$ such that:*

$$\Pr\left(\left|\|\hat{x} - \hat{y}\|_2^2 - 2(1 - \text{CosineSimilarity}(x, y))\right| \geq \varepsilon\right) \leq 6\exp\left(-\frac{k\varepsilon^2}{C'}\right) \tag{17}$$

*Proof.* We seek to bound the deviation of the squared distance. Recall the identity:

$$\left|\|\hat{x} - \hat{y}\|_2^2 - 2(1 - \text{CosineSimilarity}(x, y))\right| = 2\left|\hat{x}^T\hat{y} - \text{CosineSimilarity}(x, y)\right| \tag{18}$$

For brevity, denote $\rho = \text{CosineSimilarity}(x, y)$. Notice that following 2 definitions, we have

$$\|u\|_2 = \sqrt{k \cdot S_{uu}}, \quad \|v\|_2 = \sqrt{k \cdot S_{vv}}, \quad u^T v = k \cdot S_{uv} \tag{19}$$

Thus

$$\hat{x}^T\hat{y} = \frac{u^T v}{\|u\|_2\|v\|_2} = \frac{kS_{uv}}{\sqrt{kS_{uu}}\sqrt{kS_{vv}}} = \frac{kS_{uv}}{k\sqrt{S_{uu}S_{vv}}} = \frac{S_{uv}}{\sqrt{S_{uu}S_{vv}}} \triangleq \hat{\rho} \tag{20}$$

Following Eq. 18, it suffices to prove that the probability of the event $\{2|\hat{\rho} - \rho| \geq \varepsilon\}$ is small.

*The "Good" Event.* Define a specific tolerance $\delta = \frac{\varepsilon}{10} < 0.1$, and the "good" event $\mathcal{G}$ as the intersection of three conditions where the sample averages (from Lemma 2) are close to their expectations:

$$\mathcal{G} = \{|S_{uu} - 1| \leq \delta\} \cap \{|S_{vv} - 1| \leq \delta\} \cap \{|S_{uv} - \rho| \leq \delta\} \tag{21}$$

*Deterministic Bound under $\mathcal{G}$.* We now show that if $\mathcal{G}$ holds, the distance error is strictly less than $\varepsilon$. We analyze the error term $E = |\hat{\rho} - \rho|$:

$$E = \left|\frac{S_{uv}}{\sqrt{S_{uu}S_{vv}}} - \rho\right| = \frac{|S_{uv} - \rho\sqrt{S_{uu}S_{vv}}|}{\sqrt{S_{uu}S_{vv}}} \tag{22}$$

Since $\delta = \frac{\varepsilon}{10} < 0.1$, we have lower bounds for the denominators: Under $\mathcal{G}$, $S_{uu} \geq 1 - \delta$ and $S_{vv} \geq 1 - \delta$. Thus,

$$\sqrt{S_{uu}S_{vv}} \geq 1 - \delta > 0.9 \tag{23}$$

Using the triangle inequality, we bound the numerator:

$$\left|S_{uv} - \rho\sqrt{S_{uu}S_{vv}}\right| = \left|(S_{uv} - \rho) + \rho(1 - \sqrt{S_{uu}S_{vv}})\right| \leq |S_{uv} - \rho| + |\rho| \cdot \left|1 - \sqrt{S_{uu}S_{vv}}\right| \tag{24}$$

From $\mathcal{G}$, we know $|S_{uv} - \rho| \le \delta$. For the second term, we use the algebraic inequality $|\sqrt{ab} - 1| \le |ab - 1|$ (valid for products near 1).

$$
\begin{aligned}
|\sqrt{S_{uu}S_{vv}} - 1| &\le |S_{uu}S_{vv} - 1| \\
&= |S_{uu}(S_{vv} - 1) + (S_{uu} - 1)| \\
&\le |S_{uu}||S_{vv} - 1| + |S_{uu} - 1| \\
&\le (1 + \delta)\delta + \delta \\
&= 2\delta + \delta^2 \\
&\le 2.1\delta \quad (\delta^2 \le 0.1)
\end{aligned}
\tag{25}
$$

Substituting this back into the numerator bound (and noting $|\rho| \le 1$):

$$
\left| S_{uv} - \rho\sqrt{S_{uu}S_{vv}} \right| \le |S_{uv} - \rho| + |\rho| \cdot \left| 1 - \sqrt{S_{uu}S_{vv}} \right| \le \delta + 1 \cdot (2.1\delta) = 3.1\delta
\tag{26}
$$

Finally, we bound the total error $E$ following Eq. 23, 26:

$$
E \le \frac{3.1\delta}{0.9} < 4\delta
\tag{27}
$$

Recall we are bounding the distance deviation,

$$
\begin{aligned}
\left| \|\hat{x} - \hat{y}\|_2^2 - 2(1 - \text{CosineSimilarity}(x,y)) \right| &= 2\left| \hat{x}^T\hat{y} - \text{CosineSimilarity}(x,y) \right| \\
&= 2E \le 8\delta = 0.8\varepsilon < \varepsilon
\end{aligned}
\tag{28}
$$

Thus, the event $\mathcal{G}$ strictly implies that the squared distance deviation is less than $\varepsilon$.

*The Complement Event.* It follows that under the complement event $\mathcal{G}^c$, where the squared distance deviatation is at least $\varepsilon$ is

$$
\mathcal{G}^c = \overline{\{|S_{uu} - 1| \le \delta\} \cap \{|S_{vv} - 1| \le \delta\} \cap \{|S_{uv} - \rho| \le \delta\}}
\tag{29}
$$

$$
= \overline{\{|S_{uu} - 1| \le \delta\}} \cup \overline{\{|S_{vv} - 1| \le \delta\}} \cup \overline{\{|S_{uv} - \rho| \le \delta\}}
\tag{30}
$$

$$
= \{|S_{uu} - 1| > \delta\} \cup \{|S_{vv} - 1| > \delta\} \cup \{|S_{uv} - \rho| > \delta\}
\tag{31}
$$

Using the Union Bound and Lemma 2, the probability of failure ($\mathcal{G}^c$) is bounded by the sum of the individual failure probabilities:

$$
\Pr\left( \left| \|\hat{x} - \hat{y}\|_2^2 - 2(1 - \text{CosineSimilarity}(x,y)) \right| \ge \varepsilon \right) = \Pr(\mathcal{G}^c)
\tag{32}
$$

$$
\le 2e^{-k\delta^2/8} + 2e^{-k\delta^2/8} + 2e^{-k\delta^2/C}
\tag{33}
$$

$$
\le 6\exp\left( -\frac{k\varepsilon^2}{C'} \right)
\tag{34}
$$

$$
\tag{35}
$$

where $C'$ absorbs the constants and the factor of $(1/10)^2$. This completes the proof. $\qquad\square$

Equipped with the lemmas above, we can now prove the main theorem.

**Theorem 1** (EveryGraph's Unified Representation Space *Provably* Preserves Cosine Similarity)**.** *Let* $x, y \in \mathbb{R}^d$ *that are sampled from a set* $\mathcal{X}$ *of* $N$ *samples, and let* $A \in \mathbb{R}^{k \times d}$ *be a random matrix with entries* $A_{ij} \sim \mathcal{N}(0,1)$. *Denote the* $L_2$ *normalized projections* $\hat{x} = \frac{Ax}{\|Ax\|_2}$, $\hat{y} = \frac{Ay}{\|Ay\|_2}$. *Set the error tolerance* $\varepsilon > 0$ *and the probability of failure* $\delta \in (0,1)$. *Thus for* $k = O\left( \frac{\ln N}{\varepsilon^2} \right)$,

$$
\Pr\left( \sup_{x,y \in \mathcal{X}} \left\{ \left| \|\hat{x} - \hat{y}\|_2^2 - 2\left( 1 - \text{CosineSimilarity}(x,y) \right) \right| \right\} \ge \varepsilon \right) \le \delta
\tag{36}
$$

*Proof.* First, observe that the normalized projection is scale-invariant. For any scalar $c > 0$:

$$\frac{A(cx)}{\|A(cx)\|_2} = \frac{cAx}{|c|\|Ax\|_2} = \frac{Ax}{\|Ax\|_2} \tag{37}$$

Therefore, without loss of generality, we assume for the remainder of the proof that all inputs $x, y \in \mathcal{X}$ are unit-norm vectors, i.e., $\|x\|_2 = \|y\|_2 = 1$. Consequently, $\text{CosineSimilarity}(x,y) = x^T y$.

We aim to bound the deviation of the squared Euclidean distance in the projected space for all pairs in the dataset $\mathcal{X}$. Using Lemma 3, the probability of this event is bounded by:

$$\Pr\left(\left|\|\hat{x} - \hat{y}\|_2^2 - 2(1 - \text{CosineSimilarity}(x,y))\right| \geq \varepsilon\right) \leq 6\exp\left(-\frac{k\varepsilon^2}{C}\right) \tag{38}$$

There are $N$ samples, so there are $\binom{N}{2} < N^2$ pairs. Applying the Union Bound:

$$\Pr\left(\sup_{x,y \in \mathcal{X}} \left|\|\hat{x} - \hat{y}\|_2^2 - 2(1 - \text{CosineSimilarity}(x,y))\right| \geq \varepsilon\right) \tag{39}$$

$$\leq \sum_{x,y} \Pr\left(\left|\|\hat{x} - \hat{y}\|_2^2 - 2(1 - \text{CosineSimilarity}(x,y))\right| \geq \varepsilon\right) \tag{40}$$

$$\leq N^2 \cdot 2\exp\left(-\frac{k\varepsilon^2}{C}\right) \tag{41}$$

Setting this total probability to be at most $\delta$:

$$2N^2 \exp\left(-\frac{k\varepsilon^2}{C}\right) \leq \delta$$

$$\implies \ln(2N^2) - \frac{k\varepsilon^2}{C} \leq \ln(\delta)$$

$$\implies \frac{k\varepsilon^2}{C} \geq \ln(2N^2) + \ln(1/\delta)$$

$$\implies k \geq \frac{C}{\varepsilon^2}\left(2\ln N + \ln(2/\delta)\right)$$

To conclude the proof, we express $k$ in asymptotic notation with respect to $N$ and $\varepsilon$:

$$k = O\left(\frac{\ln N}{\varepsilon^2}\right) \tag{42}$$

$\square$

To conclude this theory discussion, we restate Corollary 1, which follows Theorem 1.

**Corollary** (EveryGraph's Unified Representation Space Preserves Cosine Similarity)**.** *Let $x, y \in \mathbb{R}^d$ and let $A \in \mathbb{R}^{k \times d}$ be a random matrix with entries $A_{ij} \sim \mathcal{N}(0,1)$. Denote the $L_2$ normalized projections $\hat{x} = \frac{Ax}{\|Ax\|}$, $\hat{y} = \frac{Ay}{\|Ay\|}$. Then for sufficiently large $k$,*

$$\|\hat{x} - \hat{y}\|_2^2 \approx 2(1 - CosineSimilarity(x,y))$$

*with high probability.*

# B   Architecture - Key Components

## B.1   Graph Unet

EveryGraph uses Graph UNet as it's core architecture. In this section we delve deeper into Graph UNet as a model and why we chose it as our trainable model.

**Hierarchical Representation Learning**   Graph UNet introduces graph pooling (downsampling) and unpooling (upsampling) Gao & Ji (2019) layers, mirroring the UNet architecture from image processing Ronneberger et al. (2015). This is especially important in the context of foundation models, which thrive when they can learn representations at multiple resolutions (from local neighborhoods to global context).

Further more, Graph UNet's hierarchical encoder–decoder also allows it to capture both fine-grained local patterns and global graph structure in a single pass, unlike plain message-passing GNNs (e.g., GCN, GAT) which only aggregate within a fixed number of hops Jiang et al. (2025).

Graph pooling itself, is data driven and task-agnostic, letting the model automatically learn which nodes are important for downstream tasks. This adaptability is key for foundation models, which need to generalize across varied graphs without handcrafted pooling strategies. By compressing the graph and then expanding it, Graph UNet allows information to travel over shorter effective paths in the coarsened graph. Since pooling is based on learned importance scores rather than fixed topology, Graph UNet can adapt to unseen graph structures Liu et al. (2023); Pham et al. (2025).

These characteristics are vital for a foundation model intended to generalize to entirely new domains and graph types.

## B.2   Link Prediction as a Self-Supervised Learning Mechanism for Foundation Models

Many SSL methods in GNNs (e.g., contrastive node embeddings, graph masking You et al. (2020b); Hu et al. (2020)) focus on node features or global graph properties. Link prediction directly forces the model to learn and preserve the underlying relational structure Zhou et al. (2020). This is critical for a foundation model, expected to generalize across domains by building a universal representation of connectivity patterns, independent of node semantics.

Link prediction is naturally domain-agnostic, meaning edges in molecules, social networks, knowledge graphs, transportation networks, etc., all follow structural patterns that can be learned You et al. (2020b); Hassani & Khasahmadi (2020). This makes link prediction pretraining inherently transferable to many downstream tasks without a task-specific adaptation, aligning with the foundation model philosophy.

Unlike node classification or graph property prediction, link prediction generates an abundant of training examples by creating positive (existing) and negative (non-existing) edges from the same graph, which enriches the data quantity.

Link prediction naturally combines topological clues, such as graph structure, with feature similarity cues like node attributes, rather than treating them separately. Thus, a foundation model pretrained this way learns joint embeddings that can generalize to both purely structural and feature-rich settings.

## C  Computational Complexity of the Closed-Form Adaptation

A central practical advantage of EveryGraph is that adaptation to a new dataset or task requires no gradient-based training: it is performed by a single closed-form ridge-regression solve (Section 3.3). Here we characterize the time and memory cost of that step and how it scales.

**Setup.**  Let $n$ be the number of training nodes (or graphs, for graph-level tasks), $l$ the dimension of the frozen EveryGraph feature representation $\check{X} \in \mathbb{R}^{n \times l}$, and $c$ the number of classes. The ridge solution of Equation 6 is

$$\hat{\beta} = \left( \check{X}^\top \check{X} + \lambda I \right)^{-1} \check{X}^\top Y, \qquad \check{X} \in \mathbb{R}^{n \times l},\ Y \in \mathbb{R}^{n \times c}.$$

**Time complexity.**  The solve decomposes into: forming the Gram matrix $\check{X}^\top \check{X}$ in $O(nl^2)$; forming $\check{X}^\top Y$ in $O(nlc)$; and solving the $l \times l$ regularized linear system in $O(l^3)$ (via a Cholesky factorization of the symmetric positive-definite matrix $\check{X}^\top \check{X} + \lambda I$). The total cost is therefore

$$O\left(nl^2 + l^3 + nlc\right).$$

Since the feature dimension $l$ is fixed by the architecture and independent of the dataset, and $c \ll l$ in all our benchmarks, the dominant term is $O(nl^2)$: the adaptation scales *linearly* in the number of training points $n$, with no iterative optimization. Inference on the test split is a single matrix multiplication $\hat{Y}_{\text{test}} = \check{X}_{\text{test}} \hat{\beta}$, costing $O(n_{\text{test}}\, lc)$.

**Memory complexity.**  The solve materializes the Gram matrix and its factorization, both $l \times l$, giving $O(l^2)$ working memory beyond the $O(nl)$ needed to store the features themselves. Crucially, the $O(l^2)$ term is independent of $n$: because the normal-equations form aggregates the data into the fixed-size $\check{X}^\top \check{X}$, the solver's footprint does not grow with the dataset, and $\check{X}^\top \check{X}$ and $\check{X}^\top Y$ can be accumulated in streaming fashion over mini-batches when $n$ is large.

**Comparison.**  This profile contrasts sharply with gradient-based adaptation. Fine-tuning a backbone (or training an MLP head) for $E$ epochs incurs $O(E)$ passes over the data with backpropagation through the network, plus optimizer state proportional to the number of trainable parameters, and requires hyperparameter and early-stopping choices per dataset. EveryGraph's adaptation replaces all of this with one factorization: it consumes task labels only through the closed-form solve, has a single hyperparameter $\lambda$ (selected on validation), and updates no network weights. This is what makes fine-tuning-free, in-context adaptation practical at scale, and it is not comparable in cost to methods that re-train a tokenizer and/or a full transformer per dataset.

**Empirical comparison: closed-form vs. gradient descent on the same objective.**  The asymptotic analysis is best complemented by direct measurements. Because the ridge objective is strongly convex for $\lambda > 0$, it admits a unique minimizer, so a sufficiently converged gradient-based solver on the *identical* objective (same frozen features, squared loss, $\lambda$, target encoding, intercept treatment, and split) reaches essentially the same solution and predictive accuracy; the value of the closed form is thus not higher accuracy but obtaining that solution exactly and directly. Table 10 confirms this: where gradient descent converges (Citeseer, WkCS), accuracy matches within noise, while the closed-form solve is one to two orders of magnitude faster, uses less peak memory, and requires a single step rather than hundreds to thousands of iterations, with no learning rate, epoch budget, or early-stopping criterion to tune per dataset. On Arxiv, gradient descent did not converge within a 20,000-iteration cap and trailed the exact solution by $\sim$2.5 points, a concrete instance of the convergence and tuning risk the closed-form route removes by construction.

Table 10: Closed-form ridge vs. gradient descent on the *same* ridge objective and frozen features (node classification). Accuracy in %; wall-clock in seconds; peak memory in MB.

| Dataset | Method | Accuracy (%) $\uparrow$ | Wall-clock (s) $\downarrow$ | Peak mem. (MB) $\downarrow$ | Iterations $\downarrow$ |
|---|---|---|---|---|---|
| Arxiv | Gradient-based | $61.81_{\pm 0.21}$ | $39.80_{\pm 0.28}$ | 2516 | 20000 (cap, not conv.) |
| | Closed-form (ours) | $\mathbf{64.36_{\pm 0.08}}$ | $\mathbf{0.61_{\pm 0.01}}$ | **2443** | **1** |
| Citeseer | Gradient-based | $\mathbf{67.33_{\pm 0.98}}$ | $0.0355_{\pm 0.0008}$ | 124 | $123_{\pm 5}$ |
| | Closed-form (ours) | $66.37_{\pm 1.05}$ | $\mathbf{0.0045_{\pm 0.0004}}$ | **92** | **1** |
| WkCS | Gradient-based | $72.68_{\pm 0.64}$ | $1.084_{\pm 0.068}$ | 230 | $3630_{\pm 250}$ |
| | Closed-form (ours) | $\mathbf{73.31_{\pm 0.42}}$ | $\mathbf{0.0057_{\pm 0.0004}}$ | **198** | **1** |

# D    Comprehensive Evaluation of EveryGraph

In this section, we present a more detailed evaluation of EveryGraph. Tables  11 and  12 report the performance of EveryGraph, pretrained on various datasets, across three tasks: node classification, link prediction, and graph classification. Furthermore, Table 3 extends the comparisons made in the main paper by including additional evaluation datasets and a broader set of comparable methods.

## D.1    EveryGraph Performance on Various Datasets

Table 11: EveryGraph node classification accuracy (mean $\pm$ std). Rows sorted approximately by average performance across training datasets.

| Training dataset $\rightarrow$ 
 Test Dataset $\downarrow$ | CoPhysics | Cora | Pubmed |
|---|---|---|---|
| CoPhysics | $94.56_{\pm0.05}$ | $94.54_{\pm0.06}$ | $94.52_{\pm0.08}$ |
| CoCS | $90.14_{\pm0.27}$ | $90.08_{\pm0.12}$ | $90.05_{\pm0.13}$ |
| AmzPhoto | $87.88_{\pm0.21}$ | $87.82_{\pm0.19}$ | $89.49_{\pm0.01}$ |
| Cora | $79.02_{\pm0.53}$ | $80.42_{\pm0.34}$ | $79.22_{\pm0.73}$ |
| AmzComp | $78.26_{\pm0.63}$ | $78.14_{\pm0.70}$ | $78.37_{\pm0.16}$ |
| WkCS | $73.19_{\pm0.27}$ | $73.34_{\pm0.33}$ | $73.03_{\pm2.09}$ |
| Pubmed | $76.68_{\pm1.25}$ | $76.68_{\pm0.88}$ | $77.49_{\pm0.73}$ |
| Citeseer | $66.98_{\pm1.06}$ | $67.48_{\pm0.56}$ | $53.56_{\pm3.71}$ |
| Wiki | $60.52_{\pm1.36}$ | $60.53_{\pm1.09}$ | $61.55_{\pm1.65}$ |
| FCora | $54.46_{\pm0.48}$ | $54.57_{\pm0.43}$ | $54.37_{\pm0.62}$ |
| Arxiv | $64.33_{\pm0.20}$ | $64.39_{\pm0.14}$ | $64.44_{\pm0.17}$ |
| Chameleon | $47.59_{\pm1.60}$ | $49.45_{\pm1.62}$ | $48.92_{\pm1.73}$ |
| AirUS | $39.75_{\pm1.37}$ | $39.59_{\pm2.91}$ | $39.79_{\pm0.23}$ |
| AirBrazil | $25.76_{\pm1.55}$ | $24.96_{\pm1.17}$ | $25.48_{\pm4.89}$ |
| Squirrel | $36.45_{\pm0.70}$ | $37.55_{\pm0.62}$ | $36.16_{\pm3.44}$ |
| AirEU | $27.10_{\pm1.65}$ | $27.52_{\pm2.83}$ | $26.53_{\pm1.66}$ |

## D.2    Comparing EveryGraph to Other Methods

Table 13 reports the performance of EveryGraph alongside a range of additional node-level baselines. These methods were selected to provide a broad and fair comparison across the datasets considered. For each dataset, we evaluate EveryGraph against commonly used node classification approaches, highlighting both traditional graph neural architectures and more recent state-of-the-art techniques.

## D.3    Random Projection VS PCA

In this subsection we compare the proposed random projection against Principal Component Analysis (PCA) as the mechanism for constructing the unified representation space.

**Why random projection.**    Let $k$ be the embedding dimension of our model and $d$ the input feature dimension. Both PCA and a random Gaussian projection are rank-$\min(d, k)$ maps, and both are lossy dimensionality reductions when $d > k$; the full-rank property therefore does not, on its own, distinguish the two. The decisive distinction is *data dependence*: PCA must be re-fit on every dataset, since its projection is estimated from that dataset's feature covariance, whereas the random Gaussian projection is a single, *data-independent* map that carries a distribution-free preservation guarantee (Theorem 1) holding for any input without re-estimation. This is precisely the property required by a task-agnostic model that must operate over a heterogeneous collection of datasets: a shared PCA basis would be capped by the dataset with the fewest features, forcing the representation dimension down to the smallest common denominator, while the random projection maps any $d$ to a fixed $k$ with no such floor. For the case $k > d$, PCA can be

Table 12: EveryGraph link prediction ROC-AUC (mean ± std). Rows sorted approximately by average performance across training datasets.

| Training dataset → 
 Test Dataset ↓ | CoPhysics | Cora | Pubmed |
|---|---|---|---|
| CoPhysics | $0.88_{\pm0.00}$ | $0.94_{\pm0.01}$ | $0.85_{\pm0.00}$ |
| Pubmed | $0.91_{\pm0.01}$ | $0.93_{\pm0.01}$ | $0.90_{\pm0.00}$ |
| Citeseer | $0.89_{\pm0.02}$ | $0.90_{\pm0.01}$ | $0.91_{\pm0.01}$ |
| CoCS | $0.91_{\pm0.01}$ | $0.93_{\pm0.01}$ | $0.94_{\pm0.00}$ |
| Cora | $0.89_{\pm0.01}$ | $0.87_{\pm0.02}$ | $0.90_{\pm0.01}$ |
| Arxiv | $0.84_{\pm0.01}$ | $0.85_{\pm0.01}$ | $0.85_{\pm0.01}$ |
| FCora | $0.85_{\pm0.02}$ | $0.87_{\pm0.02}$ | $0.86_{\pm0.01}$ |
| Wiki | $0.91_{\pm0.00}$ | $0.92_{\pm0.01}$ | $0.92_{\pm0.00}$ |
| WkCS | $0.80_{\pm0.00}$ | $0.80_{\pm0.00}$ | $0.80_{\pm0.00}$ |
| Chameleon | $0.82_{\pm0.06}$ | $0.86_{\pm0.05}$ | $0.86_{\pm0.03}$ |
| AmzComp | $0.82_{\pm0.00}$ | $0.81_{\pm0.01}$ | $0.81_{\pm0.01}$ |
| Squirrel | $0.79_{\pm0.02}$ | $0.81_{\pm0.04}$ | $0.81_{\pm0.01}$ |
| AmzPhoto | $0.78_{\pm0.01}$ | $0.77_{\pm0.00}$ | $0.77_{\pm0.00}$ |
| AirEU | $0.74_{\pm0.01}$ | $0.78_{\pm0.05}$ | $0.77_{\pm0.03}$ |
| AirUS | $0.65_{\pm0.05}$ | $0.74_{\pm0.12}$ | $0.69_{\pm0.09}$ |
| AirBrazil | $0.78_{\pm0.03}$ | $0.79_{\pm0.06}$ | $0.78_{\pm0.06}$ |

Table 13: Node classification accuracy. Best in-context method on each dataset is marked in **bold**. † indicates values reported by the original authors, ‡ indicates the results provided here were taken from Wang et al. (2025)

| Category | Dataset → 
 Method ↓ | ARXIV 
 ACCURACY ↑ | CORA 
 ACCURACY ↑ | PUBMED 
 ACCURACY ↑ | CITESEER 
 ACCURACY ↑ | CoPHYSICS 
 ACCURACY ↑ | WIKI 
 ACCURACY ↑ |
|---|---|---|---|---|---|---|---|
| Supervised | MLP HUANG ET AL. (2021) | $61.75_{\pm1.85}$ | $75.77_{\pm1.47}$ | $84.30_{\pm1.39}$ | $72.45_{\pm1.51}$ | $97.34_{\pm0.45}$ | $80.32_{\pm1.62}$ |
| | GCN Kipf & Welling (2017) | $64.16_{\pm0.55}$ | $82.11_{\pm1.23}$ | $86.16_{\pm0.27}$ | $75.75_{\pm1.64}$ | $95.87_{\pm0.40}$ | $77.47_{\pm2.41}$ |
| | GAT Veličković et al. (2018) | $63.38_{\pm1.28}$ | $82.82_{\pm1.20}$ | $85.75_{\pm0.84}$ | $75.75_{\pm1.64}$ | $96.29_{\pm0.49}$ | $79.59_{\pm1.90}$ |
| | GATv2 Brody et al. (2022) | $63.86_{\pm1.23}$ | $79.94_{\pm0.58}$ | $86.33_{\pm0.40}$ | $66.04_{\pm0.68}$ | $96.06_{\pm0.14}$ | $74.61_{\pm5.82}$ |
| | GIN Xu et al. (2019) | $58.81_{\pm0.40}$ | $82.18_{\pm1.21}$ | $84.76_{\pm0.69}$ | $65.76_{\pm1.27}$ | $96.07_{\pm0.21}$ | $69.02_{\pm4.74}$ |
| | Graphormer Ying et al. (2021) | $63.98_{\pm1.23}$ | $76.24_{\pm2.71}$ | $87.43_{\pm0.54}$ | $66.04_{\pm0.68}$ | $97.05_{\pm0.17}$ | $77.41_{\pm1.43}$ |
| Finetuned | GFT† Wang et al. (2024) | $71.93_{\pm0.12}$ | $78.62_{\pm1.21}$ | $77.19_{\pm1.99}$ | – | – | – |
| | GIT-S† Wang et al. (2025) | $76.17_{\pm1.70}$ | $78.90_{\pm1.44}$ | $81.97_{\pm0.80}$ | $73.13_{\pm0.11}$ | – | – |
| | GraphMAE† Hou et al. (2022) | $71.75_{\pm0.17}$ | $84.20_{\pm0.40}$ | $81.10_{\pm0.40}$ | $73.40_{\pm0.40}$ | – | – |
| | BGRL ‡ Thakoor et al. (2021) | $67.62_{\pm0.19}$ | $71.06_{\pm2.84}$ | $68.75_{\pm3.69}$ | $80.56_{\pm1.59}$ | – | – |
| In-Context | GraphAny (Cora) Zhao et al. (2025) | $58.59_{\pm0.09}$ | $80.13_{\pm0.06}$ | $76.50_{\pm0.20}$ | $64.48_{\pm0.57}$ | $92.61_{\pm0.51}$ | $56.05_{\pm2.46}$ |
| | EveryGraph (Cora) | $64.39_{\pm0.14}$ | $\mathbf{80.42_{\pm0.34}}$ | $76.68_{\pm0.88}$ | $67.02_{\pm0.50}$ | $94.54_{\pm0.06}$ | $60.68_{\pm0.97}$ |
| | EveryGraph (CoPhysics) | $64.33_{\pm0.20}$ | $79.02_{\pm0.53}$ | $76.68_{\pm1.25}$ | $66.98_{\pm1.06}$ | $\mathbf{94.56_{\pm0.05}}$ | $60.52_{\pm1.36}$ |
| | EveryGraph (Pubmed) | $\mathbf{64.44_{\pm0.17}}$ | $79.22_{\pm0.73}$ | $\mathbf{77.49_{\pm0.73}}$ | $\mathbf{67.98_{\pm0.27}}$ | $94.52_{\pm0.08}$ | $\mathbf{61.89_{\pm1.37}}$ |

applied by zero-padding the input features to dimension $k$ (which preserves distances exactly); we include this padded variant as a baseline below. Random projection thus remains attractive as a *single uniform mechanism* that covers both the $d > k$ and $k > d$ regimes without special-casing.

For these experiments, we set $k = 1024$ as the projection dimension and use Cora as the training dataset in all experiments.

To further test this comparison across datasets of differing feature dimension, we ran the additional experiments in Table 15.

As Tables 14 and 15 show, random projection holds a clear advantage on link prediction (e.g., CoCS, Co-Physics, Pubmed, AmzPhoto) and is on par with PCA on node classification, where the two are close in either direction. We therefore characterize random projection as offering a link-prediction-specific advantage together with node-classification parity, rather than a uniform superiority. Combined with its

Table 14: Random projection vs PCA. For datasets with $d > k$, PCA is applied to the zero-padded features.

| Task | Link Prediction (ROC-AUC) ↑ | | Node Classifcation - Accuracy ↑ | |
|---|---|---|---|---|
| Test Dataset | PCA | Random Projection - ours | PCA | Random Projection - ours |
| Cora | $0.89_{\pm 0.02}$ | $0.87_{\pm 0.02}$ | $78.04_{\pm 0.93}$ | $80.42_{\pm 0.34}$ |
| CoCS | $0.81_{\pm 0.03}$ | $0.93_{\pm 0.01}$ | $91.91_{\pm 0.78}$ | $90.08_{\pm 0.12}$ |
| CoPhysics | $0.88_{\pm 0.04}$ | $0.94_{\pm 0.01}$ | $95.37_{\pm 0.12}$ | $94.56_{\pm 0.05}$ |
| Citeseer | $0.85_{\pm 0.03}$ | $0.90_{\pm 0.01}$ | $62.20_{\pm 1.29}$ | $67.02_{\pm 0.50}$ |

Table 15: Random projection vs PCA (additional datasets, PCA with zero-padding where $d < k$).

| Task | Link Prediction (ROC-AUC) ↑ | | Node Classification - Accuracy ↑ | |
|---|---|---|---|---|
| Test Dataset | PCA | Random Projection - ours | PCA | Random Projection - ours |
| Pubmed | $0.89_{\pm 0.05}$ | $\mathbf{0.93_{\pm 0.01}}$ | $71.29_{\pm 1.58}$ | $\mathbf{76.71_{\pm 0.39}}$ |
| AirBrazil | $0.80_{\pm 0.08}$ | $0.80_{\pm 0.07}$ | $24.80_{\pm 1.13}$ | $24.80_{\pm 1.13}$ |
| AmzPhoto | $0.54_{\pm 0.01}$ | $\mathbf{0.78_{\pm 0.01}}$ | $\mathbf{90.21_{\pm 0.37}}$ | $87.94_{\pm 0.09}$ |

data-independence and applicability across arbitrary feature dimensions, this makes random projection the preferable choice for a task-agnostic model.

## D.4 Effect of $L_2$ Normalization

The unified representation space of EveryGraph applies an $L_2$ normalization to the randomly projected features (Equation 1). This normalization is essential to our theory: Theorem 1 is proven specifically for the $L_2$-normalized projection, where the squared Euclidean distance in the projected space concentrates around $2(1 - \mathrm{CosineSimilarity}(x, y))$. To confirm that the normalization matters empirically and not only in the proof, we ablate it directly: we run the identical pipeline with and without the $L_2$ normalization of Equation 1, keeping all other components fixed.

Table 16: Effect of $L_2$ normalization on the random projection. Link prediction is reported as ROC-AUC and node classification as accuracy.

| Task | Link Prediction (ROC-AUC) ↑ | | Node Classification - Accuracy ↑ | |
|---|---|---|---|---|
| Test Dataset | Without $L_2$ | With $L_2$ - ours | Without $L_2$ | With $L_2$ - ours |
| Cora | $0.50_{\pm 0.00}$ | $\mathbf{0.89_{\pm 0.02}}$ | $67.64_{\pm 0.40}$ | $\mathbf{80.42_{\pm 0.34}}$ |
| Pubmed | $0.58_{\pm 0.02}$ | $\mathbf{0.94_{\pm 0.01}}$ | $72.71_{\pm 0.76}$ | $\mathbf{77.19_{\pm 0.42}}$ |
| AmzComp | $0.50_{\pm 0.00}$ | $\mathbf{0.82_{\pm 0.01}}$ | $31.06_{\pm 4.16}$ | $\mathbf{78.71_{\pm 0.48}}$ |

As Table 16 shows, the effect is large rather than marginal. Without normalization, link prediction collapses to chance (ROC-AUC $\approx 0.50$) and node classification degrades sharply (e.g., AmzComp drops from 78.71 to 31.06). This confirms that $L_2$ normalization contributes materially to downstream performance and is not merely a theoretical convenience, consistent with the role it plays in Theorem 1.

## D.5 Closed-Form Adaptation VS Learned Adapters

In this subsection we evaluate the effectiveness of the closed-form ridge regression solution offered in our framework and compare it against learned alternatives on the same frozen features. We stress that our closed-form adaptation is, concretely, a form of linear probing on a frozen backbone: the backbone is fixed once, and only the adapter is fit per downstream task. This raises a natural question, isolated below: is the benefit due to the linear inductive bias, or to the closed-form solution specifically?

To answer this, we compare adapters trained on the identical training mask and frozen features: a fine-tuned MLP (2 layers, hidden dimension 256, 20 epochs per dataset); a linear SVM, which shares the linear inductive bias of ridge regression but is trained by optimization; and our closed-form ridge regression. The fully controlled solver comparison (closed-form vs. gradient descent on the identical ridge objective, reporting accuracy, runtime, memory, and iteration counts) is given in Appendix C, Table 10.

Table 17: Adaptation methods on the same frozen EveryGraph features (node classification accuracy, %). See Table 10 for the closed-form vs. gradient-descent comparison on the identical ridge objective.

| Training dataset | Cora | Arxiv | CoPhysics | All Datasets Average |
|---|---|---|---|---|
| EveryGraph (Ridge, closed-form – ours) | $\mathbf{80.42}_{\pm\mathbf{0.34}}$ | $\mathbf{64.39}_{\pm\mathbf{0.14}}$ | $\mathbf{94.54}_{\pm\mathbf{0.06}}$ | |
| EveryGraph (MLP) | $75.40_{\pm0.28}$ | $39.57_{\pm1.82}$ | $80.89_{\pm2.76}$ | $49.87_{\pm0.44}$ |
| EveryGraph (linear SVM) | $68.00_{\pm5.57}$ | $31.45_{\pm0.58}$ | $90.26_{\pm0.52}$ | $63.24_{\pm2.22}$ |

Table 17 shows that the MLP trails substantially and that the linear SVM, despite sharing the linear inductive bias, also trails. Since ridge and the SVM differ in loss and objective, this gap alone cannot be attributed to the closed-form solve; isolating the solver requires comparing the closed-form ridge against gradient descent on the *identical* ridge objective. Because that objective is strongly convex for $\lambda > 0$, a converged gradient solver reaches essentially the same accuracy (Table 10); the closed form's benefit is therefore practical rather than in accuracy: it returns the exact minimizer in a single deterministic step, with no learning-rate, epoch, or early-stopping tuning, at substantially lower wall-clock and memory cost. This preference for a linear, closed-form estimator aligns with Yarotsky (2017), which states that in the linear case simple linear estimators achieve the same approximation error without the overhead of neural networks, and Lathuilière et al. (2018), which shows that when the task is simple and the data is scarce, linear models can match or outperform MLPs.

### D.6   Pretraining on Multiple Graphs

Our main experiments pretrain EveryGraph on a single graph at a time. A natural question is whether the method benefits from greater diversity in its pretraining data, as is typical for foundation models in other modalities. We therefore compare single-graph pretraining against pretraining on mixtures of graphs: a homophilic citation mixture (`cora+pubmed+citeseer`) and a heterogeneous air-traffic mixture (`airbrazil+airus+aireu`), evaluated under the identical in-context protocol.

Table 18: Single-graph vs. multi-graph pretraining. Link prediction is reported as ROC-AUC and node classification as accuracy.

| Test Dataset | cora | cora+pubmed+citeseer | airbrazil+airus+aireu |
|---|---|---|---|
| *Link Prediction (ROC-AUC)* ↑ | | | |
| Citeseer | **0.917** | 0.786 | 0.807 |
| CoCS | **0.939** | 0.744 | 0.840 |
| CoPhysics | **0.954** | 0.761 | 0.855 |
| *Node Classification (Accuracy)* ↑ | | | |
| Wiki | **60.82** | 59.87 | 59.61 |
| AirUS | **39.26** | 38.82 | 36.46 |
| WkCS | **73.56** | 73.35 | 73.36 |

As Table 18 shows, in the current setup naive mixture pretraining did *not* improve over a well-chosen single graph, and on link prediction it was clearly worse. We attribute this to the pretraining objective: a link-prediction loss over a mixture of homophilic and heterophilic graphs yields a conflicting training signal, since the association "edge $\Rightarrow$ feature similarity" holds under homophily but is inverted under heterophily. Consequently, naive concatenation of datasets does not yet reconcile their differing structural regularities.

We report this as a limitation and view principled multi-graph pretraining, for example homophily-aware routing or grouping, as a promising direction for future work.

### D.7 Fine-tuning EveryGraph

We fine-tuned our model by extending training from the original checkpoints: 20 additional epochs for link prediction, and 20 more using CE-loss for node classification, all initialized from GraphUNET pre-trained weights. Model checkpoint is taken from training on the Cora dataset. As can be seen in Table 19, fine-tuning improves the results, but the added computational cost conflicts with our main objective.

Table 19: Fine Tuning EveryGraph

| Task | Link Prediction (ROC-AUC) $\uparrow$ | | Node Classifcation - Accuracy $\uparrow$ | |
|---|---|---|---|---|
| Test Dataset | no fine tuning | fine tuned | no fine tuning | fine tuned |
| Wiki | $0.92_{\pm 0.01}$ | $0.87_{\pm 0.02}$ | $60.68_{\pm 0.97}$ | $60.48_{\pm 1.16}$ |
| Pubmed | $0.93_{\pm 0.01}$ | $0.82_{\pm 0.03}$ | $76.68_{\pm 0.88}$ | $77.50_{\pm 1.17}$ |
| AmzComp | $0.81_{\pm 0.01}$ | $0.85_{\pm 0.00}$ | $78.89_{\pm 0.78}$ | $82.04_{\pm 0.18}$ |
| AirUS | $0.74_{\pm 0.12}$ | $0.87_{\pm 0.02}$ | $39.59_{\pm 2.91}$ | $39.00_{\pm 1.48}$ |

The backbone ablation, the sensitivity analysis of the projection dimension $k$, and the in-domain graph-classification transfer are presented and discussed in the main text (Section 4), as they bear directly on the central claims of the paper.

## E Datasets Statistics

In this section, we provide basic statistics about the datasets used for training and evaluation in this work. Node and link datasets are found in Table 20, while graph-level datasets are listed in Table 21.

Table 20: Basic statistics of node classification and link prediction datasets used in this work

| Dataset | #Nodes | #Edges | Node Features | #Classes | Domain |
|---|---|---|---|---|---|
| arxiv | 169 343 | 1 166 243 | 128 | 40 | citation |
| cora | 2 708 | 10 556 | 1 433 | 7 | citation |
| pubmed | 19 717 | 88 651 | 500 | 3 | citation |
| citeseer | 3 327 | 9 228 | 3 703 | 6 | citation |
| CoCS | 18 333 | 81 894 | 6 805 | 15 | collaboration |
| CoPhysics | 34 493 | 247 962 | 8 415 | 5 | collaboration |
| vessel | 3 538 495 | 5 345 897 | 3 | n/a | biology |
| airbrazil | 132 | 1 074 | 132 | 4 | transportation |
| airUS | 1 190 | 13 599 | 1 190 | 4 | transportation |
| airEU | 400 | 5 995 | 400 | 4 | transportation |
| AmazonComp | 13 381 | 245 778 | 767 | 10 | product |
| AmazonPhoto | 7 487 | 119 043 | 745 | 8 | product |
| Chameleon | 2 277 | 36 101 | 2 325 | 5 | wikipedia |
| Squirrel | 5 201 | 217 073 | 2 089 | 5 | wikipedia |
| FCora | 19 793 | 126 842 | 8 710 | 70 | citation |
| Wiki | 2 405 | 17 981 | 4 973 | 17 | wikipedia |
| WkCS | 11 701 | 431 726 | 300 | 19 | wikipedia |

Table 21: Basic statistics of graph classification datasets used in this work

| Dataset | # Graphs | Avg. #Nodes | Avg. #Edges | # Classes | Domain |
|---|---|---|---|---|---|
| MUTAG | 188 | 17.93 | 19.79 | 2 | chemical |
| PROTEINS | 1 113 | 39.06 | 72.82 | 2 | protein |
| NCI1 | 4 110 | 29.87 | 32.30 | 2 | chemical |
| NCI109 | 4 127 | 29.68 | 32.13 | 2 | chemical |

## F    Fully-Supervised Methods Training Parameters

In Section 4, we compared EveryGraph's performance with several *fully-supervised* GNNs.

In our experiment we used the same hyperparameters and training settings for all pairs of GNN and dataset. We used two layers and cross entropy was employed as the loss function in all models. Specifically, binary cross entropy with logits was used for link prediction. As these models were trained on each dataset in a fully-supervised manner, the **input and output dimensions** were determined by the dataset. For all datasets, the number of node features was used as the input dimension and the number of classes was used as output dimension.

Table 22: Hyperparameters used in fully-supervised GNN training. The figures reported here were found using a hyperparameter search procedure. * For the `vessel` dataset only, we used `batch_size = 4096`.

| Sampler | Hidden Dim. | Batch Size | Depth | Dropout |
|---|---|---|---|---|
| GraphSAINT | 1024 | 512* | 2 | 0.3 |
| **Optimizer** | **LR** | **Decay** | **Epochs** | **Patience** |
| AdamW | 9.45e−5 | 1e−2 | 2000 | 100 |

## G    EveryGraph Learning Setting and Training Parameters

In Section 3 we stated that EveryGraph utilizes a GraphUNet model. This is the only trainable component in EveryGraph and we now lay out the training and model parameters. A summary of these details is found in Table 23.

In all the experiments reported in this paper, we used the GraphUNet implementation from py-torch_geometric[1]. The GraphUNet module has depth of 2 layers. We use 1024 as the input, hidden, and output dimensions. The model is trained for 2000 epochs using AdamW optimizer, while we employ early-stopping with `patience=100 epochs`. We use binary cross entropy as our loss function (recall that the GraphUNet module is performing link predictions). To handle large graphs, we use the GraphSAINT Zeng et al. (2020) node sampler. We reiterate that as stated in Section 3, we use a random projection that maps the features of any input dataset to the fixed-size input space discussed above.

Table 23: Link Prediction GraphUNet training hyperparameters. The figures reported here were found using a hyperparameter search procedure. * For the `vessel` dataset only, we used `batch_size = 4096`.

| Input Dim. | Hidden Dim. | Output Dim. | Batch Size | Depth | Dropout |
|---|---|---|---|---|---|
| 1024 | 1024 | 1024 | 512* | 2 | 0.1487 |
| **Sampler** | **Optimizer** | **LR** | **Decay** | **Epochs** | **Patience** |
| GraphSAINT | AdamW | 9.45e−5 | 1e−2 | 2000 | 100 |

---

[1]https://pytorch-geometric.readthedocs.io/en/latest/generated/torch_geometric.nn.models.GraphUNet.html

