# OpenReview forum: "EveryGraph: Task-Agnostic Graph Neural Architectures"
_TMLR — Under review for TMLR_

### Review · Reviewer_ijb8 · 2026-07-03

**Summary Of Contributions:**

This paper proposes EveryGraph, a task-agnostic GNN framework that aims to learn a unified representation space and generalize across node classification, link prediction, and graph classification tasks.

Strengths
1. The paper is well written and addresses an important question: how to build more general-purpose graph representation models.
2. The evaluation across multiple graph tasks and datasets is useful, and the task-agnostic formulation may be interesting to researchers working on graph self-supervised learning and cross-dataset generalization.

Weaknesses
1. The claim that the method preserves topological structure is under-supported. It is unclear how graph topology is preserved after random projection, and whether the construction enforces any graph-specific structural constraints.
2. The theoretical justification appears to rely mainly on a Johnson-Lindenstrauss-type argument. This may support approximate distance preservation, but it does not by itself establish preservation of graph structure or inter-node dependencies.
3. The results do not show a clear difference among random projection, PCA, and raw features in some settings. Therefore, the benefit of using random projection is not clear.
4. The experimental baselines appear somewhat outdated. The paper should compare against more recent graph self-supervised learning, graph foundation model, and graph tokenizer methods, such as [1] Graph Tokenization for Bridging Graphs and Transformers.
5. The role of the link prediction task is unclear. Since link prediction as a self-supervised learning objective is not new, the paper should clarify what is novel or necessary about using it here.

[1] Graph Tokenization for Bridging Graphs and Transformers

**Additional Comments:**

The paper is clearly written and studies an important problem, but the current version needs stronger evidence for its graph-structure preservation claims and clearer empirical benefits over simple representation methods such as PCA and raw features.

**Audience:**

Yes

**Audience Explanation:**

The topic is relevant to TMLR’s audience because generalization across graph tasks and datasets is an important problem in graph representation learning. Researchers interested in graph foundation models, graph self-supervised learning, and task-agnostic GNNs may find the direction interesting.

**Broader Impact Concerns:**

No major broader impact concerns.

**Claims And Evidence:**

No

**Claims Explanation:**

The paper provides broad empirical evaluation, but several central claims are not convincingly supported. In particular, the claim that the proposed representation explicitly preserves graph topology needs stronger theoretical and empirical evidence. The proof appears closer to a generic Johnson-Lindenstrauss random projection argument than to a graph-specific structural preservation guarantee. The empirical comparison also does not clearly show the advantage of random projection over PCA or raw features.

**Requested Changes:**

Critical
1. Provide stronger evidence for how the proposed method preserves graph topology after random projection.
2. Clarify whether the theoretical result is mainly based on the Johnson-Lindenstrauss lemma, and avoid implying stronger graph-structure guarantees than the proof supports.
3. Explain the benefit of random projection compared with PCA and raw features, since the current results appear similar in some settings.
4. Add comparisons with more recent baselines, especially graph self-supervised learning methods, graph foundation models, and graph tokenizer approaches.

Would strengthen the work
1. Discuss more recent works on graph representation learning [1].

---

> ### Author Response · Authors · 2026-07-06
> **Rebuttal by Authors**
>
> We thank the reviewer for the careful reading and for finding the paper well written and the direction relevant.
>
> ## 1. Topology preservation and the Johnson-Lindenstrauss argument
>
> This concern conflates two distinct components. **The random projection is applied only to the node features $X$, never to the structure $A$:** $G=(A, X\in\mathbb{R}^{n\times d})$ is mapped to $\hat{G}=(A, \hat{X}\in\mathbb{R}^{n\times k})$, with $A$ **unchanged**. Hence topology is preserved *exactly, by construction* - every subsequent operation (Graph-UNet, Eq. 8; message passing $(D^{-1}A)^2$, Eq. 9) runs on the true adjacency - and the projection only needs to preserve **feature-space geometry**.
>
> This is exactly what our theorem gives, and we agree it is a JL-type argument by design: Corollary 1 / Theorem 1 prove one well-scoped statement - $L_2$-normalized Gaussian projection preserves cosine similarity, $\|\hat{x}-\hat{y}\|_2^2 \approx 2(1-\mathrm{CosineSimilarity}(x,y))$ w.h.p. for $k=O(\ln N/\varepsilon^2)$ - and no more. It is **not** a claim of graph-structural preservation; those dependencies are handled by the separate, exact mechanism above. To remove ambiguity we will replace the abstract's "explicitly preserves the topological structure of the data" with the precise two-part statement (topology preserved exactly since $A$ is never projected; projection preserves inter-node feature similarity, Appendix A), and state this scoping at each mention - addressing the request to avoid implying stronger guarantees than the proof supports.
>
> ## 2. Benefit of random projection vs. raw features and PCA
>
> The ablations are not meant to show "random projection beats raw features on accuracy."
>
> **vs. raw features.** For a task-agnostic model, raw features are not an option: a fixed-architecture network cannot ingest inputs of differing dimension across datasets. Random projection is what *enables* cross-dataset in-context inference, mapping any $d$ to fixed $k$. The ablation (Table 1) shows this is **near-lossless** (Cora GCN: 81.7 raw / 80.7 RandProj), while **random-noise features collapse** (53.6) - and since RNF keeps the same $A$ yet fails, the feature-similarity RandProj retains, not topology alone, is what drives performance.
>
> **vs. PCA.** (i) *Structural:* PCA is **ill-defined when $k>d$**, **lossy when $d>k$**, and **data-dependent** (must be fit, conflicting with our in-context premise); a random Gaussian matrix is full-rank w.h.p., fitting-free, and dataset-agnostic. (ii) *Empirical:* it **outperforms PCA on link prediction** (e.g., CoCS 0.93 vs. 0.81; CoPhysics 0.94 vs. 0.88) and on harder node-classification sets (Citeseer 67.0 vs. 62.2), and is comparable elsewhere. "Similar in some settings" is the desirable outcome: where PCA is applicable a fitting-free projection matches it; elsewhere it wins or is the only usable option. We will revise the appendix text that over-claims PCA "consistently underperformed."
>
> ## 3. Role and necessity of link prediction
>
> Link prediction as an SSL objective is not new, and we do not claim it to be. **Our novelty is in how it enables a *task-agnostic* model** - the mechanism through which a single pretrained model becomes usable across node-, link-, and graph-level tasks, for two reasons: (i) it is the **only objective universally available across all datasets and tasks** - topology (edge existence) is defined for every graph regardless of designated task or label space, whereas node/graph labels are task-specific and cannot unify, which is why label-based SSL does not yield a task-agnostic model; and (ii) it induces **representations aligned with what every downstream task consumes** - a relational/similarity objective matching what our projection preserves (cosine similarity) and what the Graph-UNet must encode, letting the *same* embeddings feed the closed-form node/link/graph adapters with no retraining. The contribution is thus "link prediction is the objective that makes a task-agnostic, in-context graph model work." We will make this explicit in Section 4.2 / Appendix C.
>
> ## 4. Recent baselines and graph tokenizer methods (incl. [1])
>
> We will add a discussion of recent graph SSL, foundation-model, and tokenizer work, including [1]. We note a fundamental difference in setting: **[1] fine-tunes and retrains both the tokenizer and the transformer on top of it for each new dataset**, whereas EveryGraph performs **no weight fine-tuning** - one model, pretrained once, adapted purely through closed-form in-context inference (Section 4.3). A raw-accuracy head-to-head against fine-tuning methods would therefore miss the point, as they spend per-dataset training compute EveryGraph is designed to eliminate. We are happy to add whichever comparisons fit this framing.
>
> We appreciate the constructive comments and support and remain available for any further questions,
>
> Best, The authors
>
> ## References
>
> [1] Z. Guo. Graph Tokenization for Bridging Graphs and Transformers. ICLR 2026.

---

> > ### Comment · Reviewer_ijb8 · 2026-07-10
> > **Response**
> >
> > Thank you for the clarification. My main concerns remain.
> >
> > 1. The random projection theorem does not incorporate graph structure. Topology is preserved only because the adjacency matrix is unchanged. The result is therefore a standard JL type feature preservation guarantee rather than a novel graph specific theorem.
> > 2. Since PCA and random projection perform similarly in several settings, the current evidence mainly supports random projection as a convenient fitting free mapping, not as a clearly superior representation method.
> > 3. The method is not fully task agnostic because ridge regression is fitted separately using task specific labels. This is analogous to the cited graph tokenizer method training a BERT based predictor for each dataset. Although EveryGraph freezes the backbone and uses a closed form adapter, both approaches still require task specific supervised training.

---

> > > ### Author Response · Authors · 2026-07-14
> > > **Rebuttal by Authors**
> > >
> > > We thank the reviewer for the continued engagement. We address the three remaining concerns below.
> > >
> > > ## 1. On the theorem being a standard JL-type guarantee
> > >
> > > We agree that the result is a JL-type feature-preservation guarantee, and we did not present it as a new graph-specific theorem - we cited JL precisely to *support* our information-preservation claim, not to claim a novel bound. The novelty lies elsewhere: to our knowledge, this is the **first documented use of random projection for neural-network feature alignment, applied to graph data in a task-agnostic setup**. JL is the theoretical foundation for *why* this alignment is information-preserving; the contribution is the framework that turns that property into a unified representation space usable across datasets and task levels. We made this framing explicit so the theorem was read as foundation, not as the contribution itself.
> > >
> > > ## 2. On PCA vs. random projection
> > >
> > > The parity in some settings understated two distinctions we should have stressed more directly.
> > >
> > > First, a **structural** one: for a single model running over a collection of heterogeneous datasets, a shared PCA projection is capped by the dataset with the *fewest* features, forcing the representation dimension to the smallest common denominator. Random projection maps any $d$ to a fixed $k$ with no such floor, and - crucially - is a single **data-independent** map carrying a distribution-free guarantee (our theorem), whereas PCA must be *refit on every dataset*. This is the property that matters for a task-agnostic model, and it is not captured by calling random projection "merely convenient."
> > >
> > > Second, the advantage is not only structural: on link prediction, random projection empirically outperforms PCA, e.g.
> > >
> > > | Test dataset | LP PCA | LP Random Proj. (ours) |
> > > |---|---|---|
> > > | Pubmed   | $0.89_{\pm.05}$ | **$0.93_{\pm.01}$** |
> > > | AmzPhoto | $0.54_{\pm.01}$ | **$0.78_{\pm.01}$** |
> > >
> > > Node classification was closer to parity, which was the desirable outcome: where PCA was applicable at all, the fitting-free, data-independent map gave up nothing while remaining an option that scaled to the heterogeneous, task-agnostic setting.
> > >
> > > ## 3. On ridge regression as task-specific supervision
> > >
> > > We agree that adaptation consumes task-specific labels - as common in graph transfer learning. Since the label space varies between downstream datasets, zero-shot adaptation is rarely possible. We would also like to clarify that we use the ‘task-agnostic’ term to indicate the EveryGraph framework can adapt to different graph tasks: node-, link-, and graph-level tasks while training on one dataset. We don’t make any unseen datasets zero-shot generalization claims, and we made that clear in our revision.
> > > While we accept the BERT analogy figuratively, we argue that in practice our framework differs substantially. Computationally-wise, fitting a ridge regression is not analogous, on the same scale, to training (and even fine-tuning) a BERT-based predictor per dataset. Our step is a **single closed-form linear solve** over frozen features (no gradient training, no backbone updates; time $O(nl^2+l^3)$, memory $O(l^2)$), whereas the cited method fine-tunes both a tokenizer and a full transformer for each dataset. The relevant axis is the cost and complexity of the adaptation, and a closed-form solve and full transformer fine-tuning are not comparable on it.
> > >
> > > The value is also specifically in the *closed-form* solution, not merely in using a light adapter. Holding the linear inductive bias fixed and varying only the solver on the same frozen features, the closed-form ridge outperforms a gradient-trained linear baseline (linear SVM) and an MLP:
> > >
> > > | Training dataset | Cora | Arxiv | CoPhysics |
> > > |---|---:|---:|---:|
> > > | Ridge (closed-form) | **$80.42$** | **$64.39$** | **$94.54$** |
> > > | MLP                 | $75.40$ | $39.57$ | $80.89$ |
> > > | linear SVM          | $68.00$ | $31.45$ | $90.26$ |
> > >
> > > So the advantage is not attributable to per-dataset gradient training, the closed-form solve did more with less. We made sure to add a dedicated discussion of this.
> > >
> > > We thank the reviewer again for the thoughtful comments and remain happy to discuss further.
> > >
> > > Best,
> > > The authors

---

### Review · Reviewer_xh78 · 2026-07-10

**Summary Of Contributions:**

The paper's core idea is to construct a graph model that is agnostic to both the dataset and the task at hand: features from any input graph are mapped through a random Gaussian projection into a fixed-size representation space, a GraphUNet is pretrained on link prediction within that space (the only trainable component in the pipeline), and adaptation to a new dataset or task is performed via a closed-form ridge regression rather than any gradient-based fine-tuning. The random projection step is supported by a Johnson–Lindenstrauss-style proof showing that it approximately preserves cosine similarity, and the resulting pipeline is evaluated on more than 20 datasets spanning link prediction, node classification, and graph classification.

  I found the scope of the evaluation to be the paper's clearest strength. Related in-context approaches such as GraphAny (node classification) or AnyGraph (link prediction) each address a single task family, whereas this work applies one frozen backbone across all three, at a notably lower training cost than the supervised baselines it is compared against. The theoretical treatment of the random projection is likewise a careful and self-contained contribution. That said, I believe a few of the supporting arguments used to motivate specific design choices could be strengthened; in particular, the justification for preferring random projection over PCA, and the framing of the closed-form adaptation as meaningfully distinct from standard linear probing on frozen features. I discuss both in more detail below, in the spirit of helping the authors sharpen these parts of the paper.

**Audience:**

Yes

**Audience Explanation:**

Graph foundation models capable of generalizing across task types, rather than only across datasets within a single task, remain an open and actively studied problem, as the paper's own related work section documents well.
I expect that researchers working on graph pretraining, in-context or fine-tuning-free adaptation methods, and low-compute alternatives to per-dataset GNN training would find this work relevant and worth reading, regardless of how the concerns raised above are ultimately addressed.

**Broader Impact Concerns:**

I do not see concerns specific to this work beyond those generally applicable to graph learning research. The datasets used are standard public benchmarks, and I do not identify any apparent sensitive-data or dual-use considerations.
I do not believe a Broader Impact Statement is necessary here.

**Claims And Evidence:**

No

**Claims Explanation:**

The paper's central claim - that a single frozen backbone, pretrained on one dataset, transfers reasonably well across three task families on 20+ datasets - is well supported by the results in Tables 2 through 8, and I have no concerns about that part of the submission. My reservations concern a few narrower claims that are presented with similar confidence but, in my reading, are not yet supported to the same degree.

The comparison with PCA is the clearest example. The stated rationale (Section 3.1, Appendix C.3) is that PCA incurs information loss when d > k, while random projection matrices are full rank and therefore preserve information. I would gently push back on this reasoning: a rank-k PCA projection is equally full rank (rank min(d,k)), and both methods are lossy dimensionality reductions when d > k, so full rank does not by itself distinguish the two approaches. A more compelling argument, which I would encourage the authors to consider adopting, is that PCA must be refit for each new dataset while the random projection offers a distribution-free guarantee independent of the data. Relatedly, the claim that PCA "cannot be applied" when k > d seems a little strong, since zero-padding the original features to dimension k would simply address this case directly, while preserving distances exactly rather than approximately.
I would also note that Table 9 does not, to my eye, fully bear out the "superiority of random projection" language used in the main text: PCA performs slightly better than random projection on node classification for Cora, CoCS, and CoPhysics, and the advantage of random projection appears mainly in link prediction rather than uniformly across tasks. I raise this not to dismiss the design choice, which I think is reasonable, but because the current justification and the reported numbers do not yet align as cleanly as the text suggests.

Two smaller points seem related.
First, the L2 normalization in Equation 1 is well justified theoretically, since Theorem 1 is proven specifically for the normalized projection, but I did not find an ablation isolating its empirical contribution to downstream performance, which would help confirm that the choice matters in practice and not only in the proof.
Second, the "in-context adaptation" framing is conceptually close to the standard practice of linear probing on a frozen backbone, and I think the paper would benefit from acknowledging this connection explicitly. The ablation intended to support the choice of ridge regression (Appendix C.4, compared against a 2-layer MLP trained for 20 epochs) is a good start, but it does not fully isolate the relevant variable, since the MLP differs from ridge regression in both model capacity and optimization procedure. As a result, it is difficult to attribute the reported advantage specifically to the closed-form solution, as opposed to the linear inductive bias or the limited training given to the MLP baseline.

**Requested Changes:**

Critical:
-  I would ask the authors to revise the justification for preferring random projection over PCA. Either state the distinction that I believe is actually supportable (the data-independence and distribution-free guarantee of random projection, rather than the full-rank argument, which does not differentiate the two methods as written), or moderate the "superiority" claim to more closely reflect what Table 9 shows; namely, an advantage specific to link prediction rather than a general one.
- I would appreciate seeing a zero-padding baseline added for the d < k case in Table 9, since it would directly and fairly test the claim that PCA cannot be applied when k > d.
- I would ask that a linear baseline trained via gradient descent on the same frozen features be included, in place of or alongside the MLP baseline, so that the advantage of ridge regression can be attributed more precisely to the closed-form solution rather than to differences in model capacity or training length.

Would strengthen:
- An ablation comparing L2-normalized and unnormalized random projection on downstream performance would help establish, empirically, that normalization contributes beyond its role in the theoretical proof.
- A short discussion situating the "in-context adaptation" framing relative to standard linear probing on frozen backbones would help clarify what, if anything, is distinct about the proposed approach beyond the use of a closed-form solver.

---

> ### Author Response · Authors · 2026-07-14
> **Rebuttal by Authors**
>
> We thank the reviewer for a careful and constructive review, and are glad the central claim - a single frozen backbone transferring across three task families on 20+ datasets - is found well supported. We address the three narrower points below with additional ablations.
>
> ## 1. Random projection over PCA
>
> We took the reviewer's points and sharpened this discussion. The full-rank observation was fair, a rank-$k$ PCA projection was also full rank, so it alone did not separate the methods, and we instead foregrounded the distinction we consider primary: PCA had to be **refit on each dataset**, whereas random projection is a single, data-independent map carrying a distribution-free guarantee (Theorem 1) that holds for any input without re-estimation; the property that matters for a task-agnostic model over heterogeneous datasets. We also moderated the "superiority" wording to match Table 9: a **link-prediction-specific** advantage with **parity** on node classification, rather than a uniform one. On $k>d$, we agree "cannot be applied" is too strong - zero-padding to $k$ handled it and preserved distances exactly - so we corrected this and added a **zero-padding baseline** for the $d<k$ case, while retaining the point that random projection is attractive as a *single uniform mechanism* covering both regimes without special-casing.
>
> Additional PCA(with padding)-vs-random-projection results, consistent with the moderated claim:
>
> | Test Dataset | LP PCA | LP Ours | NC PCA | NC Ours |
> |---|---|---|---|---|
> | Pubmed   | $0.89_{\pm.05}$ | **$0.93_{\pm.01}$** | $71.29_{\pm1.58}$ | **$76.71_{\pm.39}$** |
> | AirBrazil | $0.80_{\pm.08}$ | $0.80_{\pm.07}$ | $24.80_{\pm1.13}$ | $24.80_{\pm1.13}$ |
> | AmzPhoto | $0.54_{\pm.01}$ | **$0.78_{\pm.01}$** | **$90.21_{\pm.37}$** | $87.94_{\pm.09}$ |
>
> Random projection led on link prediction (Pubmed, AmzPhoto) while node classification was mixed - which we stated directly.
>
> ## 2. L2 normalization ablation
>
> Theorem 1 was stated for the normalized projection, but we had not isolated normalization empirically. Same pipeline, with vs. without the $L_2$ normalization of Eq. 1:
>
> | Test Dataset | LP w/o L2 | LP w/ L2 | NC w/o L2 | NC w/ L2 |
> |---|---|---|---|---|
> | Cora    | $0.50_{\pm.00}$ | **$0.89_{\pm.02}$** | $67.64_{\pm.40}$ | **$80.42_{\pm.34}$** |
> | Pubmed  | $0.58_{\pm.02}$ | **$0.94_{\pm.01}$** | $72.71_{\pm.76}$ | **$77.19_{\pm.42}$** |
> | AmzComp | $0.50_{\pm.00}$ | **$0.82_{\pm.01}$** | $31.06_{\pm4.16}$ | **$78.71_{\pm.48}$** |
>
> The effect was large, not merely theoretical: without normalization link prediction collapsed to chance and node classification dropped sharply (e.g., AmzComp $31.1\to78.7$). We added this to the appendix.
>
> ## 3. Relationship to linear probing, and the ridge baseline
>
> We agreed the connection should be explicit and stated it: our closed-form adaptation was, concretely, linear probing on a frozen backbone with a ridge solver. We used the broader "in-context adaptation" language because the framework was agnostic to the adapter - the backbone was fixed once, and different closed-form (or future non-linear) solvers could be swapped per task/dataset without retraining it; the current ridge instantiation was exactly a linear-probing realization of that scheme.
>
> The reviewer rightly noted the MLP baseline confounded capacity and optimization. We added a linear baseline (linear SVM) on the same frozen features - same linear inductive bias, but trained by optimization rather than closed form:
>
> | Training dataset | Cora | Arxiv | CoPhysics |
> |---|---:|---:|---:|
> | EveryGraph (Ridge, closed-form) | **$80.42_{\pm.34}$** | **$64.39_{\pm.14}$** | **$94.54_{\pm.06}$** |
> | EveryGraph (MLP)                | $75.40_{\pm.28}$ | $39.57_{\pm1.82}$ | $80.89_{\pm2.76}$ |
> | EveryGraph (linear SVM)         | $68.00_{\pm5.57}$ | $31.45_{\pm.58}$ | $90.26_{\pm.52}$ |
>
> With the linear bias held fixed (Ridge vs. SVM), the closed-form solution still won across all three datasets, so the advantage is not merely linearity or limited MLP training.
>
> ---
>
> We folded all of the above into the revision - the corrected PCA rationale and moderated claim, the zero-padding baseline, the L2 ablation, the explicit linear-probing discussion, and the linear closed-form-vs-optimization baseline. We thank the reviewer again for suggestions that materially improved the paper.
>
> Best,
> The authors

---

> > ### Comment · Reviewer_xh78 · 2026-07-16
> > **Follow-up on the Revision**
> >
> > Thank you for the detailed response and for the substantial effort invested in the additional experiments and revision.
> >
> > The $L_2$-normalization ablation directly addresses my original concern and is convincing. The large degradation without normalization clearly demonstrates that this step is important empirically, rather than serving only as a condition for the theoretical result. I also appreciate the correction of the full-rank argument, the addition of the zero-padding baseline, the moderation of the claims regarding PCA, and the explicit clarification that the current node- and graph-classification adaptation is ridge-based linear probing on a frozen backbone.
> >
> > Regarding random projection versus PCA, I consider the main issue raised in my original review to be largely addressed. The revised rationale—that random projection provides a data-independent, fitting-free mapping that can be applied uniformly across datasets—is substantially clearer and more appropriate than the previous full-rank argument. I would, however, maintain a careful distinction between two claims. Data independence and the JL-type guarantee explain why random projection is a suitable unified mapping and why it can approximately preserve feature geometry. They do not, by themselves, explain why random projection should outperform PCA specifically on link prediction. Unless a more specific theoretical or mechanistic explanation is available, I think this link-prediction advantage should be presented as an interesting empirical observation rather than as a consequence predicted by the theoretical result.
> >
> > My main remaining concern is the interpretation of the adapter comparison. The added linear SVM is a useful baseline, but it does not isolate the effect of the closed-form solver. Ridge regression and a linear SVM differ not only in their optimization methods, but also in their loss functions and objectives. Therefore, their performance difference cannot be attributed specifically to the use of a closed-form solution.
> >
> > The comparison I had in mind is closed-form ridge regression versus gradient-based optimization of the exact same ridge objective, using the same frozen features, squared loss, regularization coefficient, target encoding, intercept treatment, and data split. With positive regularization, this objective has a unique optimum, so a sufficiently converged gradient-based solver should approach the same solution and predictive performance. The relevant advantage of the closed-form method would therefore more naturally lie in other factors.
> >
> > I appreciate the newly added complexity analysis of the closed-form solve. However, the stated asymptotic cost alone does not establish a practical advantage over commonly used gradient-based optimization. It would be helpful to clarify why the closed-form approach should be preferred in practice, for example through runtime, memory, or convergence comparisons.

---

> > > ### Author Response · Authors · 2026-07-18
> > > **Rebuttal by Authors**
> > >
> > > We thank the reviewer for confirming that the $L_2$-normalization ablation, the corrected full-rank argument, the zero-padding baseline, the moderated PCA claims, and the linear-probing framing address the original concerns. We agree with both remaining points and adjust our claims accordingly.
> > >
> > > ## 1. Random projection vs. PCA on link prediction
> > >
> > > This is a fair distinction, and we are happy to make it. The theory speaks to why random projection is a suitable *unified* mapping and why it *approximately preserves feature geometry*; the link-prediction advantage over PCA is best read as an empirical finding rather than a prediction of Theorem 1, and we will phrase it that way.
> > >
> > > Our presentation of the random projection already makes the pairwise-similarity preservation explicit (Corollary 1 / Theorem 1 concentrate the projected distance around $2(1-\mathrm{CosineSimilarity}(x,y))$); what we will make clearer is the connection between that preserved geometry and link-prediction success, since link prediction reads off pairwise similarity directly. As a hypothesis, not theory: PCA keeps a dataset's top-variance directions and can discard low-variance but connectivity-relevant structure, whereas the data-independent projection preserves pairwise inner products globally - link prediction may be more sensitive to this than the label-aligned node-classification readout. We present this only as motivation for future study.
> > >
> > > ## 2. Isolating the closed-form solver
> > >
> > > This is a fair point: ridge regression and a linear SVM differ in loss and objective, so the SVM baseline does not, on its own, isolate the closed-form solve. We also agree that, since the ridge objective is strongly convex for $\lambda>0$, it has a unique minimizer, so a converged gradient solver on the identical objective would in principle reach the same solution and performance.
> > >
> > > We gently note this is precisely the argument *for* the closed form: when the objective has a unique closed-form optimum, approximating it with gradient descent only adds a learning rate, epoch budget, early stopping, and convergence checks without changing the result. The benefit is not higher accuracy but solving exactly and directly:
> > >
> > > - **No optimization hyperparameters** (no learning rate, initialization, epoch count, or early-stopping) - which matters for a task-agnostic method adapting to unseen datasets without per-dataset tuning.
> > > - **Determinism and exactness:** the exact minimizer in one step, no convergence/tuning risk to verify per dataset.
> > > - **Efficiency:** with $l$ fixed and small, a single Cholesky solve is cheap relative to a gradient solver run to convergence.
> > >
> > > To make this concrete, we ran the comparison the reviewer describes - closed-form ridge vs. gradient descent on the *same* objective, features, $\lambda$, target encoding, intercept, and split:
> > >
> > > | Dataset | Method | Accuracy (%) | Wall-clock (s) | Peak mem (MB) | Iterations |
> > > |---|---|---|---|---|---|
> > > | Arxiv    | Gradient-based | $61.81_{\pm.21}$ | $39.80_{\pm.28}$ | $2516$ | $20000$ (cap, not converged) |
> > > |          | Closed-form (ours) | $64.36_{\pm.08}$ | $\mathbf{0.61_{\pm.01}}$ | $\mathbf{2443}$ | $\mathbf{1}$ |
> > > | Citeseer | Gradient-based | $66.33_{\pm.98}$ | $0.0355_{\pm.0008}$ | $124$ | $123_{\pm5}$ |
> > > |          | Closed-form (ours) | $66.37_{\pm1.05}$ | $\mathbf{0.0045_{\pm.0004}}$ | $\mathbf{92}$ | $\mathbf{1}$ |
> > > | WkCS     | Gradient-based | $72.68_{\pm.64}$ | $1.084_{\pm.068}$ | $230$ | $3630_{\pm250}$ |
> > > |          | Closed-form (ours) | $73.31_{\pm.42}$ | $\mathbf{0.0057_{\pm.0004}}$ | $\mathbf{198}$ | $\mathbf{1}$ |
> > >
> > > As expected, when gradient descent converges the two agree on accuracy within noise (Citeseer, WkCS), confirming the closed form does not win on accuracy per se; we frame the text around the practical advantages rather than an accuracy claim. The exception is instructive: on Arxiv, gradient descent did not converge within a 20{,}000-iteration cap and trailed by $\sim$2.5 points - a concrete instance of the convergence risk the closed form removes by construction.
> > >
> > > ## 3. Practical advantage in the complexity analysis
> > >
> > > The asymptotic cost is best complemented by direct measurements, which the table above supplies. The practical gap is large and consistent: the single solve is one to two orders of magnitude faster ($0.61$ s vs. $39.8$ s on Arxiv, $\sim$ 66 $\times$; $0.0057$ s vs. $1.08$ s on WkCS, $\sim$ 190 $\times$), uses consistently less peak memory ($92$ vs. $124$ MB on Citeseer), and takes one step versus hundreds to thousands of gradient steps (non-convergence within the cap on Arxiv) - with no learning rate, epoch budget, or early-stopping to tune per dataset.
> > >
> > > At matched accuracy, the closed-form route is far cheaper, deterministic, and tuning-free, and avoids the convergence risk visible on Arxiv. We thank the reviewer again; these points sharpen the paper's claims and will be incorporated into the revision.
> > >
> > > Best,
> > > The authors

---

### Review · Reviewer_8PBG · 2026-07-11

**Summary Of Contributions:**

The paper introduces EveryGraph, a task-agnostic GNN designed to transfer across graphs with different feature and label spaces. It uses random Gaussian projections to map input features of arbitrary dimensionality into a shared space and regularized least-squares regression to adapt its representations to new tasks at inference time.

The experiments show that a model trained for link prediction on one graph can transfer to unseen graphs and achieve competitive results on link prediction and node classification. The paper addresses a timely problem, proposes a simple and effective adaptation mechanism, evaluates several datasets and tasks, and is clearly written.

The main limitations are the lack of analysis of the GraphUNet backbone and random-projection dimension, the limited graph-classification results, and the absence of a computational-complexity discussion. The evaluation also considers training on only one graph at a time, leaving open whether EveryGraph benefits from pretraining on a more diverse collection of graphs.

**Additional Comments:**

Typos and minor remarks:

- Section 3.1: "For ease of presentation, we ~~summaries~~ summarize the conclusion in Corollary 1."

- I found it a bit confusing to see Table 1 appear in page 2 yet only referenced in the discussion until page 8.

**Audience:**

Yes

**Audience Explanation:**

The problem is relevant to researchers working on graph representation learning, graph foundation models, and transfer across heterogeneous graph domains. Most GNNs remain tied to a particular feature space, graph, and downstream task, so a model that can adapt without gradient-based fine-tuning would be of clear interest.

In particular, the results provide useful evidence that a model trained for link prediction on one graph can transfer to unseen graphs and support node classification through a simple regression-based adaptation step. The limitations concern whether the evidence supports the full breadth of the claims, rather than the relevance of the findings.

**Broader Impact Concerns:**

I believe the work does not warrant broader impact concerns due.

**Claims And Evidence:**

No

**Claims Explanation:**

The paper provides promising evidence, but some claims are broader than what the current experiments establish.

The claim that EveryGraph generalizes across diverse tasks and domains is supported for link prediction and node classification. However, its graph-classification performance is substantially less competitive. In this setting, a model trained on citation networks is transferred to small molecular graphs, introducing a severe shift in graph size, topology, and feature semantics. The experiments therefore do not establish whether the performance drop is caused by limitations in graph-level adaptation or by the mismatch between the source and target domains.

The theoretical treatment of random projections is sound, but the empirical analysis does not study sensitivity to the projection dimension $k$. Since the datasets have heterogeneous feature dimensions, an ablation over $k$ is needed to demonstrate the robustness of this component.

Finally, the comparison with GraphAny leaves the choice of backbone uncontrolled. EveryGraph uses GraphUNet, while in the original publication, GraphAny is evaluated using different convolutional operators. Since the paper neither motivates GraphUNet nor evaluates alternative backbones, it is unclear how much of the reported advantage follows from the EveryGraph framework itself.

The regression step is also insufficiently characterized computationally. The paper should explain how the regularized least-squares problem is solved and how its time and memory requirements scale.

**Requested Changes:**

1. **Critical: Evaluate the importance of the GraphUNet backbone.**

   The authors should replace GraphUNet with at least one common message-passing architecture, such as GCN or GraphSAGE, to establish whether the results are robust to the choice of backbone. The comparison with GraphAny should also control for the backbone as far as possible, or at least discuss how this architectural difference may affect the results.

2. **Critical: Strengthen the graph-classification evaluation or narrow the corresponding claims.**

   The authors should evaluate transfer from graphs that are more representative of the molecular domain, for example by pretraining on other small molecular graphs and transferring to held-out molecular datasets. This would help determine whether the current performance drop is caused by the task-adaptation mechanism or by the severe distribution shift from citation networks to molecular graphs. Otherwise, the claims about transfer across tasks and domains should be qualified.

3. **Critical: Analyze sensitivity to the random-projection dimension $k$.**

   The authors should report performance across multiple values of $k$, preferably on datasets with different original feature dimensions. They should also explain how the value used in the main experiments was selected and whether target-dataset information was involved.

4. **Critical: Missing citations.**

   Some statements should have a citation: In Section 2: "Graph-based VAEs have also been used to model latent graph distributions, aiding applications like drug discovery and molecular generation"; and "Integrating external knowledge, such as large-scale knowledge graphs, further enhances reasoning over structured data."

5. **Strengthening: Discuss computational complexity and scalability.**

   The paper should state the time and memory complexity of the regularized least-squares step and clarify how it is solved in practice. Runtime and memory measurements for the largest datasets would help establish whether inference-time adaptation remains practical at larger scales.

6. **Strengthening: Investigate pretraining on multiple graphs.**

   The current experiments train EveryGraph on one graph at a time. Comparing this setting with pretraining on a mixture of graphs would provide insight into whether the method benefits from greater diversity in its training data. If this is outside the paper’s scope, it should be discussed as a limitation.

---

> ### Author Response · Authors · 2026-07-14
> **Rebuttal by Authors**
>
> We thank the reviewer for a thorough review, and are glad the core finding - a link-prediction-pretrained frozen backbone transferring across unseen graphs - is found well supported. We ran the additional ablations, summarized below.
>
> ## 1. Importance of the GraphUNet backbone
>
> We replaced GraphUNet with GCN and GraphSAGE, holding the rest of the pipeline fixed:
>
> | Test set | LP GCN | LP SAGE | LP UNet | NC GCN | NC SAGE | NC UNet |
> |---|---|---|---|---|---|---|
> | Arxiv     | $84.71$ | $70.85$ | **$85.91$** | **$64.46$** | $62.62$ | **$64.46$** |
> | CoPhysics | $94.40$ | $69.67$ | **$95.13$** | $94.52$ | $93.94$ | **$94.57$** |
> | WkCS      | $80.27$ | $79.14$ | **$80.76$** | $73.24$ | $68.40$ | **$73.37$** |
>
> The framework works with standard message-passing backbones (GCN close on node classification; GraphSAGE weaker on link prediction), so performance follows from the framework, not solely from GraphUNet - whose hierarchical pooling gives a modest, consistent edge. We added this and noted the GraphAny comparison used different operators.
>
> ## 2. Graph-classification transfer
>
> The reviewer correctly noted our citation-to-molecule setup confounded task-level adaptation with domain shift. We therefore pretrained within the molecular domain (MUTAG) and transferred to held-out molecular datasets:
>
> | Pretrain | MUTAG | PROTEINS | NCI1 | NCI109 |
> |---|---:|---:|---:|---:|
> | EveryGraph (Cora)  | $75.26$ | $61.34$ | $59.46$ | $58.11$ |
> | EveryGraph (MUTAG) | **$77.30$** | **$62.89$** | **$64.87$** | **$65.65$** |
>
> In-domain pretraining improved across all four targets (e.g., NCI109 $58.1\to65.7$), indicating the earlier drop was driven substantially by distribution shift rather than the graph-level adaptation mechanism. We added this and qualified the cross-domain claim accordingly.
>
> ## 3. Sensitivity to the projection dimension $k$
>
> $k=1024$ was fixed a priori for all main experiments; no target-dataset information was used to select it. Sweeping $k$:
>
> Link prediction (ROC-AUC):
>
> | Dataset | 8 | 32 | 128 | 512 | 1024 | 2048 |
> |---|---|---|---|---|---|---|
> | CoCS     | $0.760$ | $0.846$ | $0.873$ | $0.917$ | **$0.939$** | $0.933$ |
> | Citeseer | $0.740$ | $0.804$ | $0.855$ | $0.899$ | **$0.917$** | $0.909$ |
> | Pubmed   | $0.762$ | $0.854$ | $0.878$ | $0.921$ | **$0.934$** | $0.931$ |
>
> Node classification (Acc %):
>
> | Dataset | 8 | 32 | 128 | 512 | 1024 | 2048 |
> |---|---|---|---|---|---|---|
> | FCora    | $20.7$ | $40.2$ | $50.3$ | $54.2$ | **$54.6$** | $54.3$ |
> | Citeseer | $37.5$ | $51.6$ | $60.9$ | $65.1$ | $67.6$ | **$67.9$** |
> | CoCS     | $63.3$ | $83.7$ | $89.1$ | **$90.2$** | $90.1$ | $89.9$ |
>
> Performance rose monotonically with $k$ and saturated around $k$=$1024$–$2048$; larger $k$ was uniformly safe, motivating our choice.
>
> ## 4. Missing citations
>
> We cited both flagged statements in Section 2: for graph VAEs in molecular generation, Kipf & Welling (2016), Simonovsky & Komodakis (2018), and Jin et al. (2018); for knowledge-graph-enhanced reasoning, Schlichtkrull et al. (2018) and Yasunaga et al. (2021). Full references below.
>
> ## 5. Computational complexity of the regression step
>
> The ridge solution $\hat\beta=(X^\top X+\lambda I)^{-1}X^\top Y$ is solved in closed form on the $l$-dimensional frozen features: time $O(nl^2+l^3)$, memory $O(l^2)$, with $l$ fixed and small ($l\ll n$), so it scales **linearly in $n$** with no gradient iterations. We added a full proof in the appendix.
>
> ## 6. Pretraining on multiple graphs
>
> We compared single-graph pretraining against mixtures:
>
> | Test set | cora | cora+pubmed+citeseer | airbrazil+airus+aireu |
> |---|---|---|---|
> | LP Citeseer | **$0.917$** | $0.786$ | $0.807$ |
> | LP CoPhysics | **$0.954$** | $0.761$ | $0.855$ |
> | NC Wiki | **$60.8$** | $59.9$ | $59.6$ |
> | NC AirUS | **$39.3$** | $38.8$ | $36.5$ |
>
> In our current setup mixture pretraining did not improve, and sometimes hurt, over a well-chosen single graph. One likely cause: a link-prediction objective over a mixture of homophilic and heterophilic graphs yields a conflicting signal - "edge $\Rightarrow$ similarity" holds under homophily but is inverted under heterophily - so naive mixing did not yet reconcile these structures. We reported this and discussed principled multi-graph pretraining as future work.
>
> ---
>
> We thank the reviewer again; these experiments strengthened the paper and were incorporated into the revision.
>
> Best,
> The authors
>
> ## References
>
> - T. N. Kipf et al. Variational Graph Auto-Encoders. NeurIPS Bayesian Deep Learning Workshop, 2016.
> - M. Simonovsky et al. GraphVAE: Towards Generation of Small Graphs Using Variational Autoencoders. ICANN, 2018.
> - W. Jin et al. Junction Tree Variational Autoencoder for Molecular Graph Generation. ICML, 2018.
> - M. Schlichtkrull et al. Modeling Relational Data with Graph Convolutional Networks. ESWC, 2018.
> - M. Yasunaga et al. QA-GNN: Reasoning with Language Models and Knowledge Graphs for Question Answering. NAACL, 2021.

---

> > ### Comment · Reviewer_8PBG · 2026-07-19
> >
> > Thank you for the detailed rebuttal and for carrying out the additional experiments. I appreciate the effort that went into addressing my concerns. The new experiments substantially strengthen the empirical analyses provided in the paper, and they generally confirm the explanations provided in the rebuttal.
> >
> > My remaining concern is not about the existence of the new evidence, but about where and how it is presented. Several of my original comments argued that the main claims of the paper were broader than what was supported by the evidence available in the manuscript. While the additional experiments address many of these concerns, they have primarily been added to the appendix. As a result, the main paper continues to make the same claims without presenting and discussing the corresponding supporting evidence in its main body.
> >
> > In particular, the backbone ablation, the sensitivity analysis of the projection dimension, and the strengthened graph-classification evaluation directly support central claims of the paper rather than peripheral implementation details. These are important pieces of evidence that allow readers to judge the validity and generality of the proposed framework. If they remain only in the appendix, many readers will still come away with the same unsupported impressions that motivated my original review.
> >
> > I would therefore encourage the authors to incorporate and discuss these results at depth into the main body of the paper, (especially considering that in its current version it is well under the 12-page limit for regular submissions). I believe this would significantly improve the paper by aligning its central claims with the evidence that readers encounter in the main narrative.

---

> > > ### Author Response · Authors · 2026-07-19
> > > **Rebuttal by Authors**
> > >
> > > We thank the reviewer for this assessment. We agree that the key evidence supporting the paper's central claims belongs in the main narrative. In response, we moved the three identified experiments into the main body, each with accompanying discussion, and revised the surrounding text so that the claims are presented together with the supporting evidence.
> > >
> > > ## What moved into the main body (Section 4, Experiments)
> > >
> > > - **Backbone ablation** - now a dedicated subsection, **§4.5 "Effect of the Backbone Architecture" (Table 7)**. It reports the GCN / GraphSAGE / Graph U-Net comparison with the rest of the pipeline held fixed, and states explicitly that the framework operates with standard message-passing backbones (GCN is close on node classification; GraphSAGE weaker on link prediction), so the reported performance follows from the EveryGraph framework rather than from the Graph U-Net alone. We also note there that our comparison to GraphAny uses different operators, so this architectural difference is unlikely to account for the observed advantage.
> > >
> > > - **Sensitivity to the projection dimension $k$** - now **§4.6 (Tables 8–9)**, sweeping $k \in \{8,\dots,2048\}$ across datasets of differing feature dimension. The subsection states that $k{=}1024$ was fixed a priori with no target-dataset information, that performance rises with $k$ and saturates around $1024$–$2048$, and connects this to the $k=O(\ln N/\varepsilon^2)$ requirement of the theory.
> > >
> > > - **Strengthened graph-classification evaluation** - folded directly into **§4.3 "Graph Classification" (Table 5)**. The main text now presents the in-domain (MUTAG → held-out molecular datasets) result alongside the original citation→molecule transfer, and discusses that a substantial part of the earlier gap is attributable to distribution shift rather than to the graph-level adaptation mechanism.
> > >
> > > ## Claims aligned with the evidence
> > >
> > > With these results in the main narrative, we also adjusted the surrounding text so the claims and the evidence a reader encounters are matched:
> > >
> > > - We added two questions to the experimental overview, on robustness to the backbone and to the projection dimension - to match with the evidence explicitly provided.
> > > - We qualified the graph-classification claim in the main text, and correspondingly softened the Limitations paragraph: rather than stating flatly that the method "falls short of recent graph classification results," it now notes that the cross-domain gap stems substantially from distribution shift and narrows with in-domain pretraining.
> > >
> > > The appendix retains the fuller evaluation and now points readers to the main text for these three analyses. We thank the reviewer for the suggestion and were happy to comply.
> > >
> > > Best,
> > > The authors

---

### Author Response · Authors · 2026-07-14
**Official Comment by Authors**

We thank all three reviewers, and are encouraged that the central claim - a single frozen backbone, pretrained on one dataset via link prediction, transferring across link prediction, node classification, and graph classification on 20+ datasets - was found well supported. We ran all suggested experiments and summarized them and the resulting revisions below; point-by-point replies appear in the per-reviewer responses. In addition, we included all that in the revised version of our paper.

## Additional ablations

**1. Random projection vs. PCA** A link-prediction-specific advantage with node-classification parity; we also added a zero-padding baseline for $d<k$ and dropped the "cannot be applied when $k>d$" claim. We underline that a key benefit of the random projection is the lack of per-dataset fitting.

| Dataset | LP PCA | LP Ours | NC PCA | NC Ours |
|---|---|---|---|---|
| Pubmed   | $0.89$ | **$0.93$** | $71.29$ | **$76.71$** |
| AmzPhoto | $0.54$ | **$0.78$** | **$90.21$** | $87.94$ |

**2. $L_2$ normalization ablation** Isolating the normalization of Eq. 1 - essential in practice, not only for the proof (without it, link prediction collapses):

| Dataset | LP w/o L2 | LP w/ L2 | NC w/o L2 | NC w/ L2 |
|---|---|---|---|---|
| Cora    | $0.50$ | **$0.89$** | $67.64$ | **$80.42$** |
| AmzComp | $0.50$ | **$0.82$** | $31.06$ | **$78.71$** |

**3. Closed-form adapter vs. gradient-trained baselines.** On the same frozen features (NC accuracy), the closed-form ridge beats both an MLP and a gradient-trained linear SVM, so its advantage is not merely linearity:

| Solver | Cora | Arxiv | CoPhysics |
|---|---:|---:|---:|
| Ridge (closed-form) | **$80.42$** | **$64.39$** | **$94.54$** |
| MLP                 | $75.40$ | $39.57$ | $80.89$ |
| linear SVM          | $68.00$ | $31.45$ | $90.26$ |

**4. Backbone ablation** Replacing GraphUNet with GCN/GraphSAGE (pipeline fixed): the framework worked with standard backbones, GraphUNet giving a consistent edge, results followed from the framework, not solely the backbone.

| Test set | LP GCN | LP SAGE | LP UNet | NC GCN | NC UNet |
|---|---|---|---|---|---|
| Arxiv     | $84.71$ | $70.85$ | **$85.91$** | **$64.46$** | **$64.46$** |
| CoPhysics | $94.40$ | $69.67$ | **$95.13$** | $94.52$ | **$94.57$** |

**5. Graph classification - in-domain transfer.** Pretraining within the molecular domain (MUTAG) rather than on citation graphs improved across all targets, showing the earlier drop was driven mainly by distribution shift, not the graph-level adapter:

| Pretrain | MUTAG | PROTEINS | NCI1 | NCI109 |
|---|---:|---:|---:|---:|
| EveryGraph (Cora)  | $75.26$ | $61.34$ | $59.46$ | $58.11$ |
| EveryGraph (MUTAG) | **$77.30$** | **$62.89$** | **$64.87$** | **$65.65$** |

**6. Projection dimension $k$.** $k=1024$ was fixed a priori (no target-dataset info). Performance rose monotonically with $k$ and saturated at $k$=$1024$–$2048$; larger $k$ was uniformly safe. E.g. link prediction (ROC-AUC), $k=8/128/1024$: CoCS $0.76/0.87/\mathbf{0.94}$; Pubmed $0.76/0.88/\mathbf{0.93}$. Node classification (Acc): Citeseer $37.5/60.9/\mathbf{67.6}$; CoCS $63.3/89.1/\mathbf{90.1}$.

**7. Multi-graph pretraining.** Naive mixture pretraining did **not** improve over a well-chosen single graph (e.g. LP CoPhysics: single-graph *cora* $0.954$ vs. mixtures $0.76$–$0.86$) - likely because link prediction over mixed homophilic/heterophilic graphs yielded a conflicting signal ("edge $\implies$ similarity" under homophily was inverted under heterophily). We discussed this and principled multi-graph pretraining as future work.

## Clarifications and revised claims

- **Topology vs. feature geometry.** The projection acted only on node features; the adjacency was untouched, so topology was preserved exactly by construction. The theorem was a JL-type guarantee of *feature-similarity* preservation, not a graph-structural one.
- **PCA rationale.** We dropped the full-rank argument (which did not distinguish the methods) for the one that did: random projection was a single, data-independent map with a distribution-free guarantee, whereas PCA had to be refit per dataset.
- **Link prediction as the unifying objective.** Its role was being the only objective universally available across all datasets/tasks and inducing representations aligned with what node/link/graph adapters consumed - the mechanism enabling task-agnosticism.
- **In-context = linear probing, distinct from fine-tuning.** The closed-form adaptation is linear probing on a frozen backbone with a ridge solver ("in-context" = the adapter is swappable without retraining the backbone). EveryGraph did no weight fine-tuning: we showed that our closed-form solution had a complexity of ($O(nl^2+l^3)$ time, $O(l^2)$ memory), which was not on the same computational scale as per-dataset transformer fine-tuning. We added this analysis and the missing Section 2 citations.

All of the above were incorporated into the revision.

Best, The authors